# Modulating multi-functional ERK complexes by covalent targeting of a recruitment site in vivo

Tamer S. Kaoud [1,2], William H. Johnson[1], Nancy D. Ebelt [1], Andrea Piserchio [3], Diana Zamora-Olivares[4], Sabrina X. Van Ravenstein [1], Jacey R. Pridgen [1], Ramakrishna Edupuganti[1], Rachel Sammons [1], Micael Cano[1], Mangalika Warthaka[1], Matthew Harger[5,11], Clint D.J. Tavares[6], Jihyun Park[7], Mohamed F. Radwan [8], Pengyu Ren [5], Eric V. Anslyn [4], Kenneth Y. Tsai [9], Ranajeet Ghose[3,10] & Kevin N. Dalby [1]*

Recently, the targeting of ERK with ATP-competitive inhibitors has emerged as a potential clinical strategy to overcome acquired resistance to BRAF and MEK inhibitor combination therapies. In this study, we investigate an alternative strategy of targeting the D-recruitment site (DRS) of ERK. The DRS is a conserved region that lies distal to the active site and mediates ERK–protein interactions. We demonstrate that the small molecule BI-78D3 binds to the DRS of ERK2 and forms a covalent adduct with a conserved cysteine residue (C159) within the pocket and disrupts signaling in vivo. BI-78D3 does not covalently modify p38MAPK, JNK or ERK5. BI-78D3 promotes apoptosis in BRAF inhibitor-naive and resistant melanoma cells containing a BRAF V600E mutation. These studies provide the basis for designing modulators of protein–protein interactions involving ERK, with the potential to impact ERK signaling dynamics and to induce cell cycle arrest and apoptosis in ERK-dependent cancers.

[1] Division of Chemical Biology and Medicinal Chemistry, The University of Texas at Austin, Austin, TX 78712, USA. [2] Department of Medicinal Chemistry, Faculty of Pharmacy, Minia University, Minia 61519, Egypt. [3] Department of Chemistry and Biochemistry, The City College of New York, New York, NY, USA. [4] Department of Chemistry, The University of Texas at Austin, Austin, TX 78712, USA. [5] Biomedical Engineering Department, The University of Texas at Austin, Austin, TX, USA. [6] Department of Cancer Biology, Dana-Farber Cancer Institute and Department of Cell Biology, Harvard Medical School, Boston, MA 02115, USA. [7] The University of Texas MD Anderson Cancer Center, Houston, TX, USA. [8] Department of Pharmaceutical Chemistry, Faculty of Pharmacy, King Abdulaziz University, Jeddah 21589, Saudi Arabia. [9] Moffitt Cancer Center, Tampa, FL, USA. [10] Graduate Programs in Biochemistry, Chemistry and Physics, The Graduate Center of the City University of New York, New York, NY 10016, USA. [11]Deceased: Matthew Harger *email: dalby@austin.utexas.edu

The MAPK ERK is a crucial regulator of cellular proliferation and survival and is activated through a core pathway comprised of a small G protein (RAS) and a sequential cascade involving three protein kinases (RAF, MEK, and ERK) (Supplementary Fig. 1a)[1,2]. ERK1 and ERK2 (referred to as ERK herein) phosphorylate a plethora of cytoplasmic proteins, including the RSKs, which along with activated ERK localize to the nucleus to promote new transcription events. Dynamics are considered to be an essential component of the ERK signaling pathway, with evidence suggesting that both the subcellular localization and duration of ERK activity determines cell fate decisions, such as proliferation[3–5]. More than 50% of melanomas, a common and deadly form of skin cancer, contain a constitutively active mutant of the kinase *BRAF* (BRAF[V600E]) that causes inappropriate ERK signaling, a dominant driver of human melanoma[6].

Within a decade of the initial discovery, the development of small molecule kinase inhibitors of BRAF (e.g., vemurafenib and dabrafenib) and their clinical validation occurred, showing significant short-term responses in patients with *BRAF*-mutant melanomas[7–9]. Clinical samples revealed several mechanisms of resistance to vemurafenib[10,11], the majority of which result in reactivation of ERK signaling[12–17]. Cotreatment with a MEK inhibitor provides a more durable suppression of ERK signaling with enhanced antitumor effects in preclinical models and, until the emergence of immunotherapy[18], was the standard of care for treating BRAF mutant melanoma[19–21]. However, even in combination with MEK inhibitors, the utility of RAF inhibitors is restricted to tumors expressing mutant BRAF and limited by the emergence of drug-resistant ERK activation[10].

Recent evidence supports the possibility that inhibitors of ERK, for example, the ATP-competitive inhibitor ulixertinib, may exhibit acceptable safety profiles[22], and can suppress the emergence of resistance and overcome acquired resistance to BRAF and MEK inhibitors[22,23]. In a series of studies, multiple clinical resistance mechanisms were suggested to converge on ERK reactivation in BRAF V600E colorectal cancer[24–26]. However, resistance to ATP-competitive ERK inhibitors is almost certain. For example, long-term exposure of cells to the ATP-competitive inhibitor of ERK, SCH772984[23] leads to acquired resistance, corresponding to a mutation, G186D, in the DFG motif of ERK1[26] and an in vitro analysis revealed other potential resistance mutations, mainly located within the ATP-binding site[27]. Therefore, complementary strategies to inhibit ERK directly, which can overcome mechanisms of resistance to ATP-competitive inhibitors of ERK, may provide significant therapeutic opportunities.

In order to form highly specific signaling complexes with components of the MAPK signaling pathway, ERK utilizes two recruitment sites, the D-recruitment site (DRS) and the F-recruitment site (FRS) that are spatially distinct from the catalytic machinery (Supplementary Fig. 1b)[28–31]. The first of these "docking" regions, the DRS, is located behind the ATP-binding pocket and recognizes partners that contain sequences, such as the D-site consensus $(R/K)_{2-3}-(X)_{2-6}-\Phi_A-X-\Phi_B$ (where $\Phi_{A/B}$ are hydrophobic residues). The second docking region, the FRS[30], is located just below the activation loop and binds substrates, such as those bearing a sequence (F-X-F-P)[32].

As the DRS is a critical mediator of numerous ERK−protein interactions[33], we reasoned that protein−protein interaction (PPI) inhibitors targeting the DRS might exhibit unique pharmacological activity, with resistance profiles orthogonal to those of ATP-competitive inhibitors.

Here we report a new modality for targeting the ERK DRS in vivo using the 1,2,4-triazol-3-one chassis, BI-78D3 (Fig. 1a), which disrupts the formation of ERK-protein complexes to affect BRAF-mutant tumor growth.

## Results

### BI-78D3 binds to the D-recruitment site of ERK.
Shapiro and colleagues identified several small molecules that exhibit dose-dependent inhibition of ERK signaling[34] and inhibition of cell proliferation in several cell lines, including melanoma cell lines[35,36]. A proposed mechanism of inhibition involved the inhibitors binding reversibly to the D-recruitment site (DRS) of ERK, to disrupt the binding of substrates, such as p90RSK. These studies provided the first proof of concept that the D-recruitment site of ERK is a potentially viable therapeutic target in cancer cell lines. However, to date, no molecules have emerged that bind this site with high enough potency to be highly effective in vivo. Several years ago, the 1,2,4-triazol-3-one, BI-78D3 was identified as a weak inhibitor of the interaction between the scaffold protein c-Jun N-terminal kinase inhibitory protein 1 (JIP1) and the c-Jun N-terminal kinase 1, JNK1[37]. We found that BI-78D3 impedes the ability of a constitutively active form of the ERK kinase, MKK1 (MKK1G7B)[38] to phosphorylate ERK1 and ERK2 in vitro and also impedes the ability of activated ERK1 and ERK2 to phosphorylate v-ets erythroblastosis virus E26 oncogene homolog 1 (Ets-1; a construct including the residues 1–138 that is necessary and sufficient for ERK-mediated phosphorylation, was used[39]) (Fig. 1b). In both cases, BI-78D3 inhibits the phosphorylation reaction in a dose-dependent manner (IC50 ~ 1.0 μM; 30 min incubation).

A plausible explanation for the efficient inhibition is that BI-78D3 impedes the formation of the MKK1G7B•ERK and ERK•Ets-1 protein complexes. To further test this possibility, we utilized solution NMR methodology taking advantage of the available assignments for backbone and methyl resonances of ERK2[40,41]. A 2D $^{15}N$, $^{1}H$ TROSY-based titration of BI-78D3 into $^{15}N$, $^{2}H$-labeled ERK2 showed significant spectral perturbations localized at, and near, the DRS. A slow exchange behavior (see below) with two sets of resonances, one corresponding to free ERK2 and a second corresponding to a BI-78D3 bound state (Supplementary Fig. 2 shows two examples, namely T108 and H123), for most of the perturbed residues were apparent. In contrast, a progressive attenuation of the C159 amide peak was noted until its complete disappearance at approximatively an equimolar (1/1) ratio (Fig. 1c) of BI-78D3. Treatment with 10 mM DTT restored the C159 resonance (Supplementary Fig. 3) and eliminated resonances corresponding to the BI-78D3-bound state for the other residues. The ability to reverse the perturbations, most notably to restore the "unbound" C159 by the use of a reducing agent, suggested that this residue was covalently modified by BI-78D3 (as described further below). The ERK2 resonances that displayed significant perturbations (>2.5σ threshold; Supplementary Fig. 4a, c (top panel); Fig. 1d, left panel) corresponded to residues that were localized in a region comprising loop 11 (N156), the spatially contiguous inter-lobe linker (T108, D109), and the αE helix (D122, H123, I124). Additionally, the analysis of perturbations induced by BI-78D3 on the methyl resonances of the Ile (δ1 only), Leu and Val residues of $^{15}N$, $^{2}H$-ILV-labeled ERK2 utilizing $^{13}C$, $^{1}H$ HMQC experiments, revealed a similar overall trend. As in the case of the amides, significant perturbations (>2.5σ threshold) included (Supplementary Fig. 4b, c (bottom panel), Fig. 1d, right panel) the δ1 methyl groups of L113 (αD) and of L119 (loop 11), both methyl groups of L155 (β7) and the δ2 methyl of L161 (β8). Taken together, these data suggest the specific localization of BI-78D3 at the ERK2 DRS. Titration of ERK2 with modified BI-78D3 ligands missing the nitro group altogether or replaced by a methyl ester (Con-1), i.e., those that are incapable of a covalent linkage with C159, showed chemical shift perturbations that were only observable at very high concentrations of ligand. This suggests that the localization of

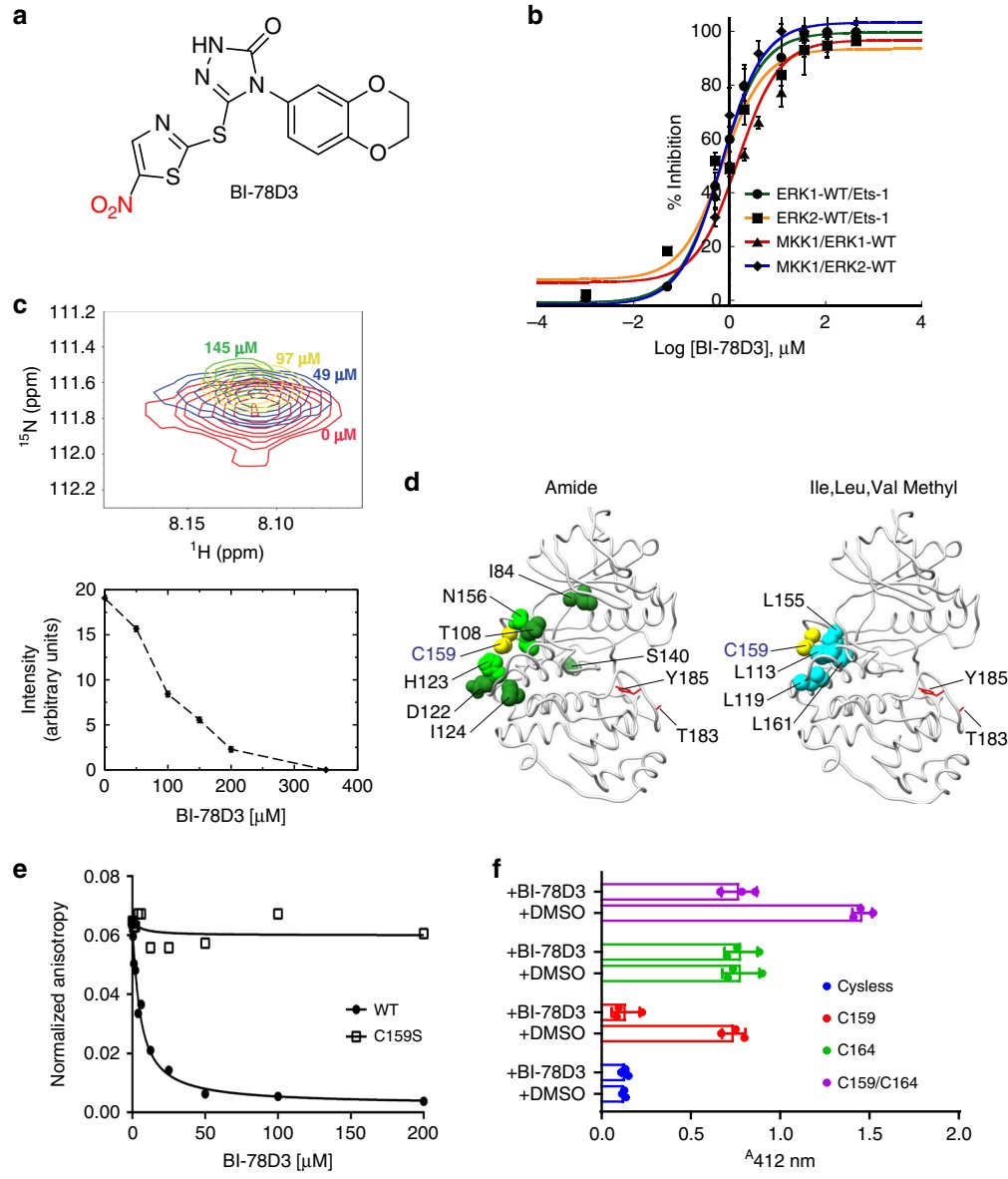

**Fig. 1** BI-78D3 labels C159 of ERK2. **a** Chemical structure of BI-78D3. **b** BI-78D3 inhibits ERK1/2 phosphorylation by constitutively active MKK1 (MKK1G7B) and inhibits the phosphorylation of Ets-1 by activated ERK1/2. Different concentrations of BI-78D3 were incubated with either unphosphorylated or activated ERK1/2 for 30 min before the addition of [γ-$^{32}$P] ATP and MKK1G7B or Ets-1, respectively (data are from three independent experiments for Ets-1 assays and two for MKK1G7B assays, and bars represent mean ± SD (standard deviation)). **c** The top panel shows an expansion of the region corresponding to C159 in $^{15}$N, $^{1}$H TROSY (600 MHz) spectra of inactive ERK2 (200 µM) in the presence of increasing amounts of BI-78D3. The bottom panel shows the change in the relative intensity of the C159 resonance in these spectra with increasing concentrations of BI-78D3. **d** Residues that show significant spectral perturbations calculated from $^{15}$N,$^{1}$H TROSY (800 MHz) or $^{13}$C, $^{1}$H HMQC (800 MHz) spectra of ERK2 in the presence of approximately equimolar amounts of BI-78D3 for backbone amide (left panel) or Ile (δ1), Val and Leu methyls (right panel) resonances are indicated on the structure of ERK2. Residues for which the amide or methyl chemical shift perturbations exceed the corresponding average plus twice the standard deviation are colored dark green and cyan, respectively. Residues for which amide resonances that are broadened to below the noise are colored light green. C159 is shown in yellow (and labeled in blue) and the activation loop T183 and Y185 are shown in red stick representation. All of the spectral perturbations in both cases are centered in and around the DRS of ERK2. **e** Fluorescence anisotropy was employed to assess the ability of BI-78D3 to competitively displace a fluorescent D-site-containing peptide from the DRS of activated ERK2 and ERK2 C159S (represents one experiment out of two repetitions). **f** Specificity of BI-78D3 towards C159 and C164. ERK proteins (5 µM) were incubated with 100 µM BI-78D3 for 60 min. Free thiol was titrated using Ellman's reagent (data are from three independent experiments, and bars represent mean ± SD)

BI-78D3 at the ERK2 DRS is primarily driven by the covalent modification of C159.

**BI-78D3 covalently modifies Cys-159.** While most of the perturbations seen for the ERK2 resonances corresponded to changes in resonance frequencies in the presence of BI-78D3, the peak

corresponding to C159 (Fig. 1c) was quantitatively attenuated, vanishing completely approximately at an equimolar ratio of the ligand. We considered the possibility that the dramatic loss of the C159 resonance was because BI-78D3 undergoes a unique interaction with C159. To test this scenario, we mutated C159 of ERK2 to Ser. In contrast to wild-type ERK2, the ERK C159S mutant was not inhibited by BI-78D3, when assayed against a

peptide substrate known to require docking within the DRS for turnover[42] (Supplementary Fig. 5). Furthermore, BI-78D3 efficiently displaced a FITC-labeled peptide containing a consensus sequence (D-site) that is known to target the DRS of ERK2. However, BI-78D3 was unable to displace the peptide from the DRS of ERK2 C159S under the same experimental conditions (Fig. 1e). The apparent ($K_i \pm$ standard error) for the binding of BI-78D3 to activated ERK2 was estimated in this fluorescence anisotropy competition assay to be $0.8 \pm 0.2 \, \mu M$ (Fig. 1e). Next, as suggested by the NMR results above, we further tested the possibility that C159 formed a covalent bond with BI-78D3 and evaluated whether preincubation of ERK2 with BI-78D3 protected C159 from reacting with 5,5′-dithio-bis-(2-nitrobenzoic acid) (Ellman's reagent). As a control, we also evaluated C164 that lies within the active site of ERK2. We prepared four forms of ERK2 containing either no cysteines (where all seven are mutated to either Ser or Ala), two cysteines (C159 and C164) or one cysteine (C159 or C164). While Ellman's reagent quantitatively titrated C164 and C159, preincubation of the proteins with 100 $\mu M$ BI-78D3 for 60 min, followed by buffer exchange, completely protected C159, but not C164 from reacting with Ellman's reagent (Fig. 1f), further suggesting that BI-78D3 indeed forms a covalent adduct with C159.

**The mechanism of addition of Cys-159 to BI-78D3.** Next, we set out to determine the structure of the adduct formed between ERK2 and BI-78D3 by first comparing its covalent modification to model reactions. Incubation of several aliphatic thiols with BI-78D3 induced bathochromic and hyperchromic shifts in the UV spectrum of BI-78D3, characterized by an isosbestic point centered around $\lambda \sim 340 \, nm$, consistent with a process involving the conversion of two spectroscopically active species (Fig. 2a and Supplementary Fig. 6a, b). Notably, incubation of BI-78D3 with ERK2 induced the same time-dependent spectroscopic changes (Fig. 2b), consistent with the notion that ERK2 undergoes a similar chemical reaction with BI-78D3 to that of model thiols. To identify the product of the reaction between ethanethiol and BI-78D3, one equivalent of ethanethiol was incubated with BI-78D3 for 1 h and the single product of the reaction purified by analytical HPLC (retention time ~60 min) (Supplementary Fig. 7a). The deconvoluted high-resolution mass spectrum of the purified product revealed a mass of 442 Da (Supplementary Fig. 7b), suggesting that BI-78D3 forms a 1:1 adduct with ethanethiol. Incubation of recombinant His-ERK2 (MW 42,318 Da) with BI-78D3 for 20 min resulted in the appearance of a new molecular weight species of 42,704 Da, corresponding to the incorporation of a single molecule of BI-78D3 (observed shift 386 Da, expected 380 Da) (Fig. 2c), supporting the notion that BI-78D3 forms a similar covalent adduct with ERK2. A covalent modification of C159 is further supported by LC-MS studies that reveal an absence of incorporation of BI-78D3 into the ERK2 C159S mutant (Supplementary Fig. 8).

To gain further insight into the mechanism of adduct formation, methoxyethanthiol was incubated with an equimolar amount of BI-78D3 in phosphate buffer (pH 7.5) for ~30 min, and the $^1H$ NMR spectrum of the product of the reaction determined (details may be found in the legend to Supplementary Fig. 9). The proton NMR spectrum of BI-78D3 is characterized by a unique downfield singlet at 8.32 ppm, which corresponds to the C4 proton of the nitrothiazole ring. Significantly, this proton is visible at 8.44 ppm in the product of the reaction between BI-78D3 and methoxyethanethiol (Supplementary Fig. 9a), suggesting that the nitrothiazole ring is not the site of thiol addition to BI-78D3, as might be expected. Instead, the $^1H$ NMR data and $^{13}C$ NMR spectrum of the product (Supplementary Fig. 9b)

suggest that the C5 carbon of the 1,2,4-triazol-3-one is the site of addition. Similar tetrahedral adducts are known in the literature[43–48]. Significantly, the purified product of the reaction between ERK2 and BI-78D3 also retains a peak at 8.32 ppm (Supplementary Fig. 10a), actively supporting the notion that ERK2 forms a product that is analogous to $T^0$ (Fig. 2d). It is noteworthy that the S-moiety of BI-78D3 provides for substantial flexibility even when BI-78D3 is covalently attached to ERK2 (through C5). It is expected that this moiety would be surface exposed allowing free rotation and resulting in a sharper proton peak than normally expected for a group attached to a macromolecule. A close inspection of this resonance (marked by the "*" in Supplementary Fig. 9 and occurring at roughly 8.32 ppm) shows a small downfield shift and broadening in the ERK2-BI-78D3 (6) spectrum compared to that in the spectrum of BI-78D3 (1) alone (full width at half maximum ~2.0 and 1.22 Hz respectively) (Supplementary Fig. 10b). A similar shift was observed for the same proton when BI-78D3 reacted with one equivalent of methoxyethanethiol (Supplementary Fig. 9a). Interestingly, further incubation of $T^0$ results in the slow appearance of a species, corresponding to the loss of 127 Da (Supplementary Fig. 11), which likely corresponds to the hydrolysis of the adduct to yield $T^1$ and 5-nitrothiazol-2(3H)-one (Fig. 2d). Thus, we propose that BI-78D3 reacts with ERK2 according to the mechanism shown in Fig. 2d.

**BI-78D3 exhibits covalent selectivity for ERK1/2.** As noted above, although ERK has seven cysteine residues, BI-78D3 reacts exclusively with C159, even after prolonged incubation. To gain insight into the kinetic mechanism of the reaction, we measured the dependence of its rate of formation on the concentration of BI-78D3, using an in vitro kinetic assay. ERK2 was incubated with different concentrations of BI-78D3 for varying amounts of time, before diluting aliquots 100-fold, at set times to determine its residual activity. Figure 3a shows a plot of the observed rate constant for inactivation of ERK2 against the concentration of BI-78D3. The line through the data corresponds to the best fit to Eq. (1) for a two-step mechanism of irreversible inhibition (Fig. 3a), with a $K_i = 2.3 \pm 0.8 \times 10^{-6} \, M$ and $k_{inact} = 1.7 \pm 0.12 \times 10^{-3} \, s^{-1}$. Thus, $k_{inact}/K_i$ which is equivalent to the second-order rate constant for the reaction of BI-78D3 with ERK2 is $7.4 \pm 2.6 \times 10^2 \, M^{-1} \, s^{-1}$ (Fig. 3a). This compares favorably with second-order rate constants of $18 \, M^{-1} \, s^{-1}$ and $7.8 \, M^{-1} \, s^{-1}$ for hydroxyethanethiol ($pKa = 9.5$) and ethanethiol ($pKa = 10.6$) respectively. Thus, the specificity of adduct formation appears to result from the formation of a reversible complex between ERK and BI-78D3, which promotes nucleophilic attack of the C159 thiolate anion to C5 of the 1,2,4-triazol-3-one ring.

To gain structural insight into the mechanism, we modeled BI-78D3 onto the surface of ERK2 (PDB: 4ERK) using a computational approach described in detail in the Methods section. Our modeling supports the idea that BI-78D3 binds in proximity to C159 and is consistent with the observed changes in the backbone chemical shifts of ERK2 upon adduct formation (Fig. 3b). However, while it is plausible that interactions with loop 11 (based on the NMR perturbations described above) are essential for orienting BI-78D3, further studies were required to assess the model. A mutational analysis that is shown in Supplementary Note 1 and Supplementary Table 1 supports the notion that prior to reacting with C159, BI-78D3 binds close to loop 11 (N156) and the spatially contiguous inter-lobe linker (T108).

Structural studies and sequence alignments (Fig. 3c) of several MAPKs reveal that the DRS is highly conserved, and a cysteine corresponding to C159 is present in all MAPKs except ERK3 and

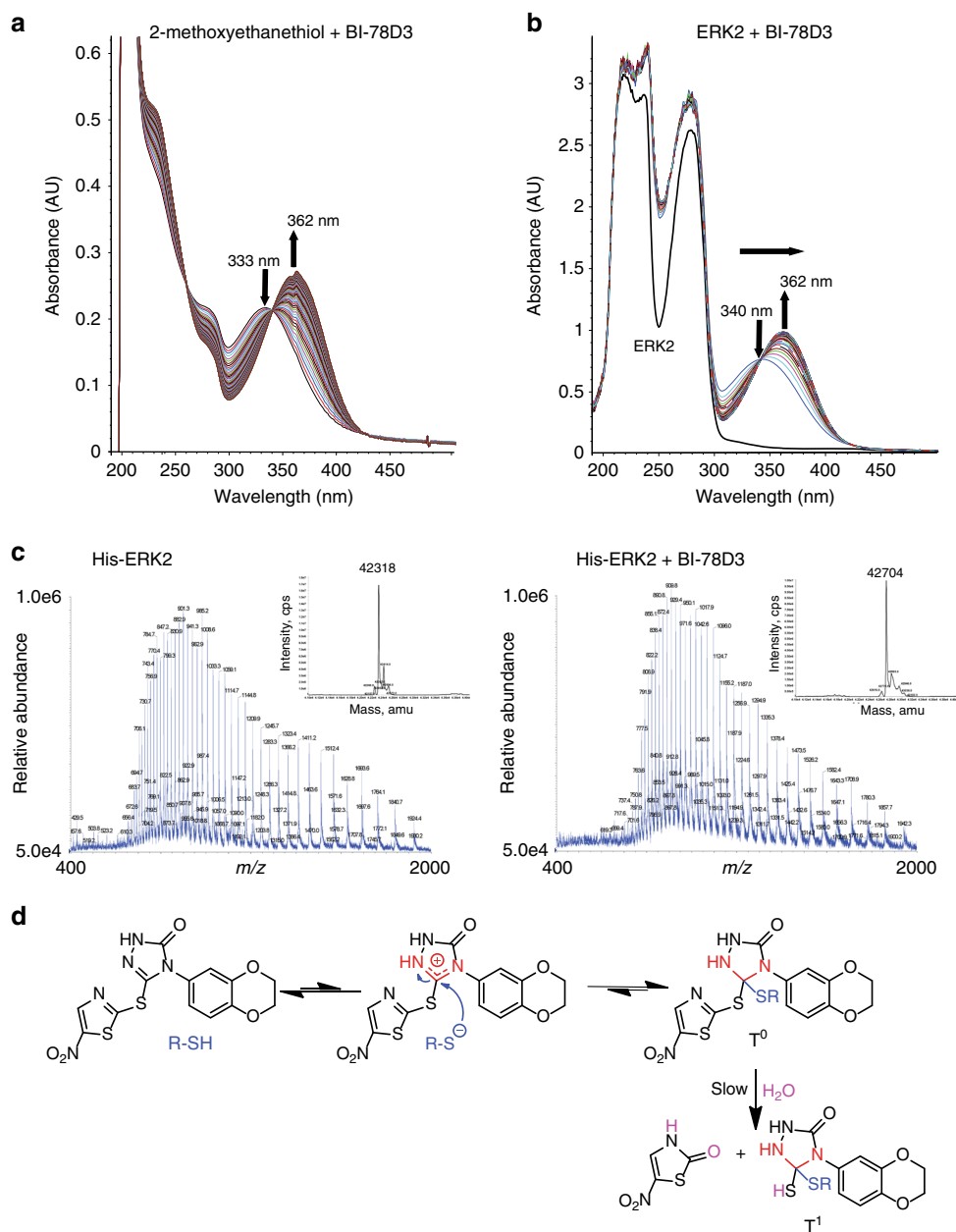

**Fig. 2** BI-78D3 reaction with free thiols afford a stable tetrahedral intermediate. **a, b** Observed change in the UV-visible spectrum for the reaction of **a** 2-methoxyethanethiol (100 μM) or **b** ERK2 (50 μM) with BI-78D3 (10 μM) in 50 mM phosphate buffer, pH 7.5 and 2% dioxane (spectra recorded every 10 s for 1000 s). **c** Deconvoluted mass spectra of ERK2 showing the addition of approximately 386 Da after incubation of ERK2 (5 μM) with BI-78D3 (100 μM) for 15–20 min, followed by buffer exchange using a PD-10 column (calculated molecular weights of BI-78D3 and ERK2 are 380 and 42,329 Da respectively). **d** Proposed mechanism of the reaction between BI-78D3 and ERK1/2

ERK4. Given this similarity, we explored the possibility that BI-78D3 might react with other MAPKs by monitoring for changes in its absorption spectrum (UV/visible). As discussed in Supplementary Note 2, among several proteins tested, only ERK2 showed a characteristic change in the absorption spectrum, consistent with thiol addition. In contrast, incubation of each protein with DNTB revealed one or more surface accessible cysteines (Supplementary Fig. 12 and Supplementary Table 2). Additionally, we could not detect the labeling of either His-JNK2, p38-α MAPK or ERK5 by BI-78D3 using LC-MS (Supplementary Fig. 13). And finally, while BI-78D3 does inhibit the JNKs in an in vitro assay (Supplementary Fig. 14), we were able to fully recover the enzymatic activity of JNK1 by dialysis following its incubation with BI-78D3 (10 μM) for 60 min (Fig. 3d).

**BI-78D3 forms a covalent adduct with ERK in mammalian cells.** We next evaluated the ability of BI-78D3 to covalently modify C159 of ERK in intact cells. HEK293 cells stably over-expressing Flag-ERK2 were incubated with BI-78D3 (25 μM) for 2 h. The cells were then lysed, and Flag-ERK2 was purified by immunoprecipitation, flash frozen to −80 °C until analyzed by LC-MS. The deconvoluted mass spectrum of transiently transfected Flag-ERK2 purified from HEK293 cells displayed three peaks corresponding to Flag-ERK2 (Fig. 4a), most likely non-phosphorylated, mono-phosphorylated, and bi-phosphorylated Flag-ERK2. Treatment of cells with BI-78D3 resulted in three new peaks (with different relative ratios), each displaying a mass shift of ~380 Da, consistent with covalent modification of ERK2 by BI-78D3 (Fig. 4a). To evaluate the pharmacodynamic properties of

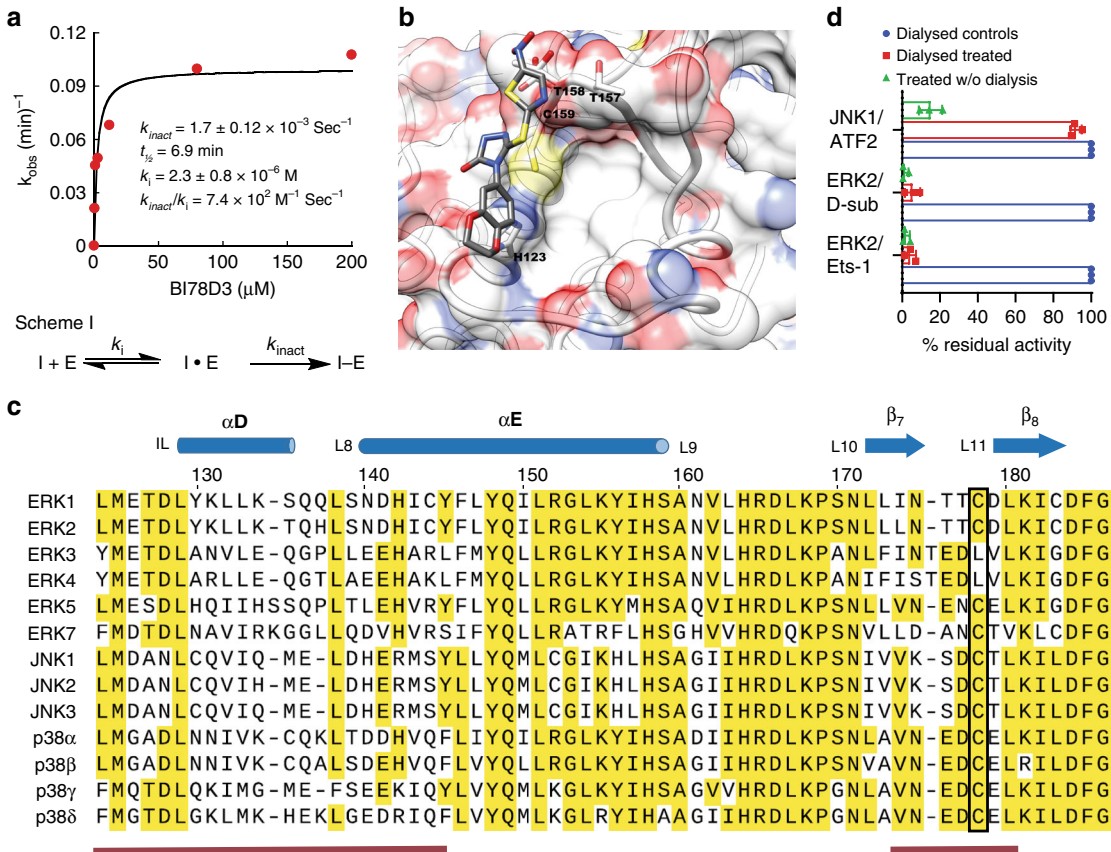

**Fig. 3** BI-78D3 exhibits a unique selectivity towards ERK1/2. **a** Kinetic behavior of ERK2 inactivation by BI-78D3. Data were fit to Eq. (1) as described in the methods section and according to the two-step mechanism of activation shown in Scheme 1 (data represents one experiment out of two repetitions; both produced consistent kinetic parameters, values of $K_i$ and $k_{inact}$ ± standard error). **b** The final frame of a 100 ns molecular dynamics trajectory for BI-78D3 binding to ERK2 (PDB 4ERK). The figure was generated using UCSF Chimera software (https://www.cgl.ucsf.edu/chimera/)[70]. **c** Sequence alignment of Human MAPKs encompassing the D-recruitment site. The cysteines corresponding to C159 in ERK2 (targeted by BI-78D3) are indicated by the black rectangle; the numbering corresponds to human ERK1 sequence. C178 in *Homo sapiens* ERK1 corresponds to C161 in *Homo sapiens* ERK2 and C159 in Rattus norvegicus ERK2. **d** Reversibility of JNK1, but not ERK2 inhibition by BI-78D3. Each enzyme (5 μM) was treated with BI-78D3 (100 μM) or DMSO (control) for 1 h. The activity of each enzyme was estimated before and after excessive dialysis (data are from three independent experiments, and bars represent mean ± SD)

BI-78D3, HEK 293 cells were incubated with 10 or 50 μM BI-78D3 for 2 h, followed by the exchange of media and the addition of EGF (30 min) at the time indicated (Fig. 4b). EGF treatment resulted in robust phosphorylation of ERK, as judged by western blotting. A single treatment with 50 μM BI-78D3 suppressed the ability of EGF to activate the ERK pathway for up to 8 h after BI-78D3 was washed out. This suggests that BI-78D3 has the potential to modify ERK for a minimum of 8 h in cells to suppress its activation. Consistent with these observations, incubation of the ERK2·BI-78D3 adduct (UV spectrum is shown in Supplementary Fig. 15a) with 5 mM glutathione for 30 min failed to rescue the activity of ERK2, as determined using an in vitro kinase assay (Supplementary Fig. 15c). Additionally, incubation of a different purified adduct (formed upon reaction of ERK2 carrying a single cysteine (C159) with BI-78D3) for 16 h at room temperature in buffer at pH 7.5 (Supplementary Fig. 15b) failed to induce reactivity with Ellman's reagent, suggesting that C159 remains protected.

**BI-78D3 suppresses ERK signaling in mammalian cells**. The A375 human melanoma cell line expresses mutant V600E BRAF and is sensitive to ERK inhibition[49]. To adopt a rigorous evaluation of the ability of BI-78D3 to covalently inhibit ERK

signaling in these cells, we adopted the following protocol: A375 cells were serum starved overnight and then incubated with BI-78D3 for 1 h, followed by washing out for 2 h and then stimulation with EGF (30 min). Using this protocol, we found that BI-78D3 exhibits a dose-dependent suppression of ERK phosphorylation within the activation loop (T183/Y185 in ERK2). Furthermore, the phosphorylation of the downstream substrate p90RSK on S359/S363 that lie on the linker between the N- and C-terminal kinase domains and T573 that is on the activation loop of the C-terminal kinase domain were also suppressed. A slight increase in the phosphorylation of MKK1 within the activation loop was noticeable, consistent with some relief of ERK-mediated feedback inhibition[50] (Fig. 4c, quantification of three different experiment is shown in Supplementary Fig. 16a).

Interestingly, a high dose of BI-78D3 (25 μM) triggered a pronounced rounding of the cells and phosphorylation of ERK, without further significant activation of MKK1/2. Notably, however, the inhibition of phospho-RSK T359/S363 was maintained (Supplementary Figs. 25 and 26a show uncropped immunoblots of three different experiments), suggesting that despite its phosphorylation ERK remained inhibited. This was confirmed using a fluorescent peptide sensor (ERK-sensor-D1)[51] to estimate the activity of ERK in cell lysates (Supplementary Fig. 26b). This suggests that higher doses of BI-78D3 may result

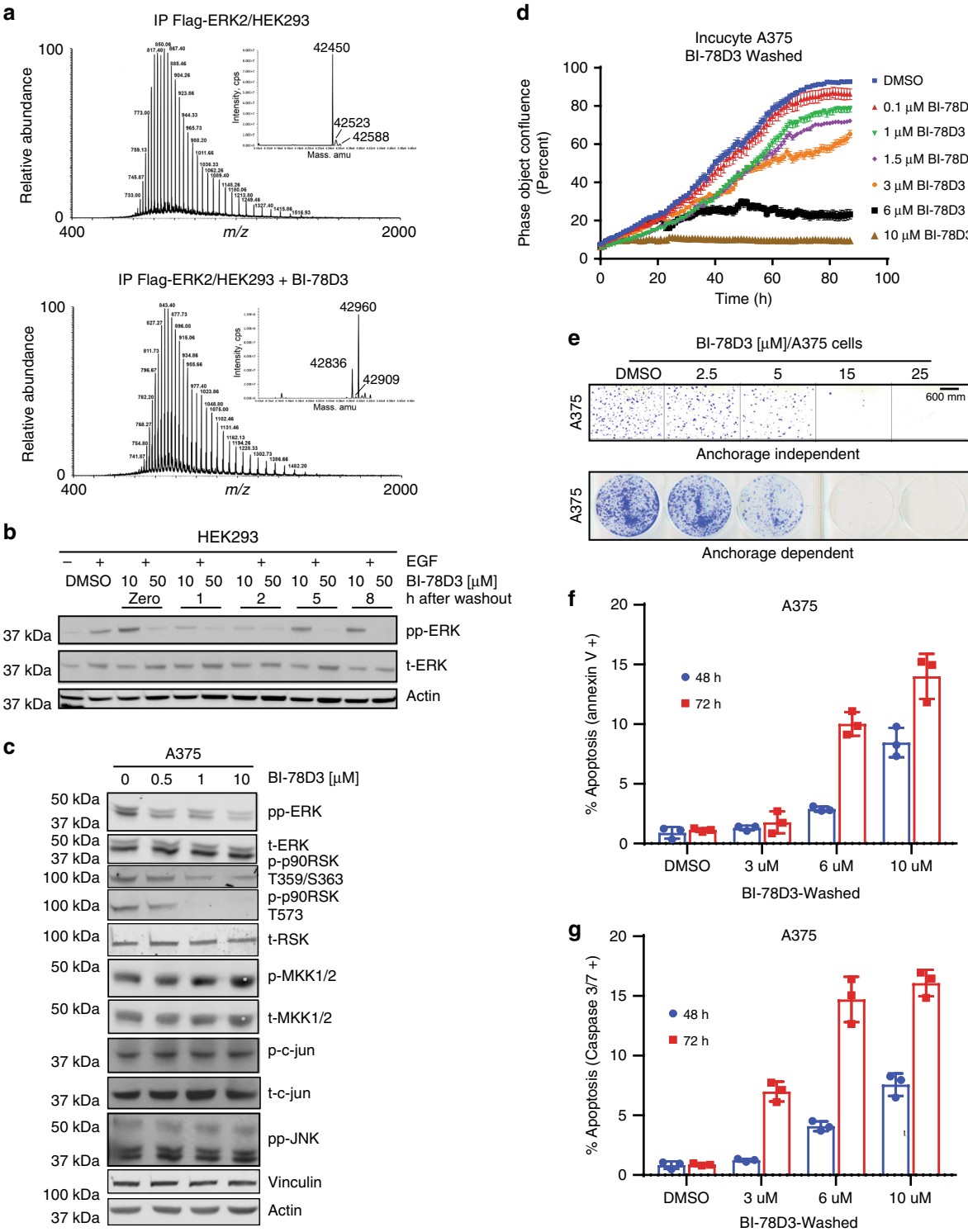

**Fig. 4** BI-78D3 labels ERK2 in cells. **a** BI-78D3 labels Flag-ERK2 covalently in HEK-293 cells. **b** Sustained inhibition of EGF-stimulated ERK phosphorylation by incubating HEK cells with BI-78D3 for 2 h, followed by washout. Immunoblots were performed in duplicate using pan-actin as a control. Images have been cropped for presentation, uncropped images are shown in Supplementary Fig. 25. **c** Inhibition of EGF-stimulated ERK activation, but not JNK activation by BI-78D3 in *BRAF*-mutant A375 melanoma cells. 1.5 million cells were seeded in 60 mm dish, serum starved overnight, treated with BI-78D3 for 1 h in serum-free media, followed by inhibitor washout for 2 h in the same media. Cells were then stimulated with EGF for 30 min in full media. This experiment was performed three times. Pan-Actin was employed as a control in each experiment. Images have been cropped for presentation, and uncropped images are shown in Supplementary Fig. 25. Two more replicates are shown in Supplementary Fig. 25b. **d** Treatment of A375 cells with BI-78D3 for 1 h in full media, followed by washout, inhibits their approach to confluence in a dose-dependent manner (IncuCyte imaging. Error bars: SEM (*n* = 3), represents one experiment of at least three repetitions). **e** BI-78D3 inhibits anchorage-dependent and anchorage-independent growth of A375 cells in a dose-dependent manner (the experiments were performed in triplicate). Scale bar represents 600 μm. **f**, **g** BI-78D3 induces apoptosis of A375 cells in a dose-dependent manner, as determined by Annexin V or Caspase-3/7 assays (IncuCyte imaging. Error bars: SEM (*n* = 3))

in the direct or indirect inhibition of a phosphatase function that dephosphorylates ERK.

Significantly, the phosphorylation of JNK, c-Jun (Fig. 4c), p38MAPK (Supplementary Fig. 16b) in A375 cells and ERK5 (Supplementary Fig. 16c) in MEL-1617 cells[15] (another *BRAF*-mutant melanoma cell line) appear to be unaffected, even when cells are subjected to treatment with high concentrations of BI-78D3. Thus, in melanoma cell lines, BI-78D3 appears to inhibit both ERK activation, as well as its downstream signaling, without significantly affecting other MAPKs.

**BI-78D3 inhibits cell proliferation**. As the ERK pathway regulates cell survival and proliferation[1], we measured the proliferation potential of cells following incubation with BI-78D3, with drug washout, by monitoring their approach to confluence. Dose-dependent suppression of cell growth ($IC_{50} \sim 3.5\,\mu M$) was noted when measured over 90 h (Fig. 4d). A similar treatment abrogated the formation of anchorage-dependent and anchorage-independent colonies (Fig. 4e). Cell cycle analysis revealed that treatment with BI-78D3 (6 μM), followed by washout for 24 and 48 h induced G1 arrest (Supplementary Fig. 17). Annexin V (Fig. 4f) and caspase3/7 assays (Fig. 4g) revealed significant dose-dependent induction of apoptosis 72 h after treatment, which is also evidenced by a significant induction of a sub-G1 population (Supplementary Fig. 18). Similar results were observed when BI-78D3 was not washed out.

The chemical genetics study shown in Supplementary Note 3 validates Cys-159 as a site of vulnerability to BI-78D3 in melanoma and HEK293 cells. When an inhibitor-resistant form of ERK2 (ERK2 C159A) was transiently expressed in A375 cells, it significantly suppressed the ability of BI-78D3 to inhibit ERK signaling (Supplementary Fig. 19), ERK nuclear localization (Supplementary Fig. 20), and the formation of anchorage-independent (Supplementary Fig. 21a) and anchorage-dependent colonies (Supplementary Fig. 21b). Similar results were seen in HEK 293 cells (Supplementary Fig. 22).

**BI-78D3 inhibits ERK to suppress melanoma cell growth in vivo**. As BI-78D3 was previously reported to have an encouraging microsome and plasma stability (half-life of approximately 54 min)[37] and had been shown to inhibit tumor growth of OVK18 xenografts that possess a naturally occurring *PIK3R1*[L370fs] mutation (10 mg kg$^{-1}$ BI-78D3 was administered intraperitoneally four times per week for 3 weeks)[52], we decided to test whether it inhibits ERK signaling in vivo. Therefore, we investigated its effect on the growth of a BRAF-mutant xenograft model established from the human melanoma cell line A375. Inbred, athymic nude mice were injected with $1 \times 10^6$ A375 cells into the right flank. When tumors reached an average volume of 150 mm$^3$, mice were randomized into two groups of ten mice each. One group was treated daily with an intraperitoneal injection of vehicle (2.5% ethanol, 5% Tween-80, 1× PBS), and the other group with 15 mg kg$^{-1}$ BI-78D3 suspended in the vehicle. Tumor volumes were measured daily until tumors reached a maximum diameter of 1 cm. The tumor measurements of these mice were analyzed for tumor growth comparison (Fig. 5a) where BI-78D3 (15 mg kg$^{-1}$ daily) caused potent tumor growth suppression after 10 days of treatment. BI-78D3 was tolerated by the mice, as measured by morbidity, lethality, and loss in body weight (Supplementary Fig. 23).

To test target engagement in the BI-78D3-treated tumors, mice were treated with vehicle or BI-78D3 at day 16 and sacrificed 3 h later. A western blot of lysate derived from fresh, frozen tissue showed that the tumor growth suppression was accompanied by robust inhibition of ERK phosphorylation, but not JNK

phosphorylation in the tumor tissues (Fig. 5b). Second, immunohistochemical staining of tumor sections for phospho-ERK confirmed the inhibition of ERK phosphorylation in the mice treated with BI-78D3 (Fig. 5c), compared to vehicle control mice. Third, using peptide-based biosensors and chemometric modeling[53], we were able to quantify the enzymatic activity of ERK in tumor lysates, and confirm the inhibition of ERK activity in the treated mice (Fig. 5d).

**BI-78D3 can overcome acquired resistance to BRAF inhibitors**. To assess whether covalent targeting of ERK by BI-78D3 can overcome acquired resistance of BRAF inhibitor-resistant cells, we used two previously reported melanoma cell lines exhibiting sensitivity (451 Lu-S and MEL 1617-S) and acquired resistance (451 Lu-R and MEL 1617-R) to SB-590885, as well as other BRAF inhibitors, such as PLX4720[15]. As expected PLX4720 has a reduced effect on cell viability (Fig. 6a, c) of both resistant cell lines. In contrast, BI-78D3 suppresses cell viability (Fig. 6b, d) and anchorage-dependent growth (Fig. 6e, f) of all four cell lines. Induction of apoptosis by BI-78D3 was observed when evaluated in the 451 cell lines, as judged by the induction of caspase3/7 activity (Fig. 6g, h). While the 451 Lu cell lines showed approximately similar sensitivity to BI-78D3 in the viability assay, the MEL1617-R cell line exhibit slightly more resistance to BI-78D3 than the MEL1617-S cell line (Fig. 6b, d). However, little difference is apparent in the colony formation assay (Fig. 6f). Interestingly, approximately tenfold more BI-78D3 is required to inhibit the ERK pathway in the MEL1617-R cells compared to the parental cells following drug washout and stimulation by EGF (Fig. 6i), suggesting that the resistant cells likely possess significantly more robust ERK signaling, which may contribute to the observed resistance. This is consistent with previous reports that MEL1617-R cells are more resistant to inhibition by the MEK inhibitor GSK1120212 than MEL1617-S cells, where a tenfold higher dose of GSK1120212 was required to inhibit ERK phosphorylation, cell viability, and G0/G1 cell cycle arrest[15].

**Discussion**

In this report, we identified BI-78D3 as a small molecule that covalently attaches to C159 of ERK and characterized the mechanism of its addition. Surprisingly, we found that C159 forms a covalent bond with the C5 carbon of the 1,2,4-triazol-3-one ring of BI-78D3 to form a relatively stable tetrahedral adduct. We hypothesize that the addition proceeds through the nucleophilic attack of the thiolate anion of C159 to the C5 carbon, promoted by the weakly acidic formamidinium ion moiety of BI-78D3 that forms upon BI-78D3 protonation (Fig. 2d). In silico predictions (Marvinview, ChemAxon) of the p$K_a$ of protonated N2 of BI-78D3 suggests a value of less than zero, suggesting that the rate of addition of C159 to such a formamidinium ion intermediate is likely to be extremely rapid, a notion supported by our previous studies on the reactivity of nucleophiles with a fluoroformamidinium ion[54].

The adduct is stable to either the reversible elimination of C159 or the expulsion of the 4-nitro-3*H*-pyrrole-2-thiolate anion, presumably as a result of the weak basicity of both nitrogen atoms of the 1,2,4-triazol-3-one ring. However, the adduct $T^0$ does react slowly over several hours, a loss of approximately 127 Da can be detected by LC-MS, which likely corresponds to hydrolysis and the loss of the 5-nitrothiazol-2(3H)-one ring to give the tertiary thiol $T^1$ (Fig. 2d). Notably, ERK remains unable to phosphorylate Ets-1, after some 13 h of incubation in buffer free of BI-78D3, suggesting that both adducts block Ets-1 binding. The absence of detectable free thiol on the ERK C159- BI-78D3 adduct after 13 h may be a result of $T^1$ not undergoing significant disulfide bond

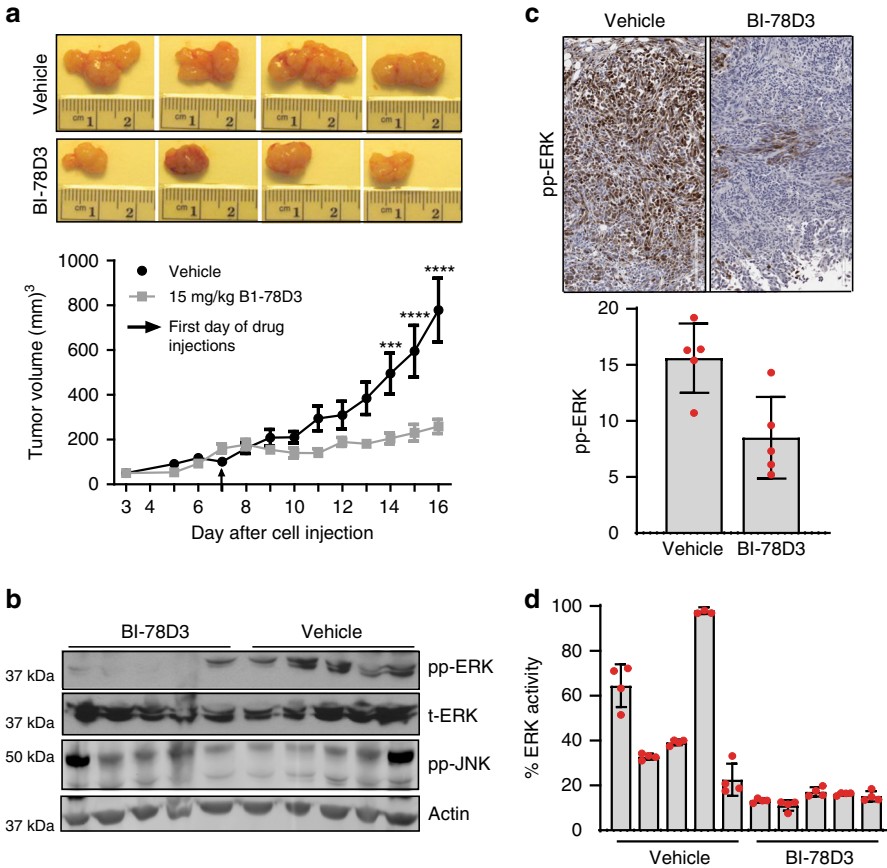

**Fig. 5** BI-78D3 inhibits ERK in human *BRAF*-mutant A375 xenograft model. **a** Female athymic nude mice were injected subcutaneously with A375 cells. Once the tumor volume reached 150 mm³, mice were randomly divided into two groups ($n = 10$) and dosed daily with vehicle or BI-78D3 (15 mg kg⁻¹ by intraperitoneal administration) for 10 days. Tumor volumes were measured daily by caliper, and the values plotted as mean ± SEM; ***$P < 0.001$; ****$P < 0.0001$; data were analyzed following a two-way repeated measures ANOVA using the Bonferroni post-test method (GraphPad Prism 7 software). **b** To confirm target engagement of BI-78D3, tumor lysates from five mice of each group were subjected to western blotting and analyzed for *pp-ERK1/2, ERK, pp-JNK*, and *β-actin* (loading control). Images are cropped for presentation; uncropped images are shown in Supplementary Fig. 25. **c** Formalin-fixed paraffin-embedded tumor sections were subjected to immunohistochemistry. Representative images of the tumor sections stained with pp-ERK are shown. Comparisons of stained cells normalized to mm² of tumor area revealed significant suppression of pp-ERK in treated mice (the values were plotted as mean ± S.D., $n = 5$ for each group). **d** Estimated amount of active ERK in tumor lysates from five mice of each group, obtained using peptide-based biosensors and chemometric modeling using calibration regression model as described in Zamora-Olivares et al.[53] (the values were plotted as mean ± SEM, $n = 4$ for each lysate)

exchange with Ellman's reagent due to its steric hindrance and low acidity[55].

It is notable that the modified C159 in ERK2 is conserved in all MAPKs except ERK3 and ERK4 (see Fig. 3c). One possible explanation for the selectivity towards this conserved cysteine in ERK1 and ERK2 is the steric constraint imposed upon the attack by a nucleophile at the C5 trigonal carbon. A central notion in organic chemistry is the well-known Bürgi–Dunitz angle, which describes the geometry of attack of a nucleophile to a trigonal center[56]. This geometrical restriction would likely preclude many surface cysteines from reacting efficiently with C5, due to the additional steric hindrance imposed by the thiazole and benzo-dioxine rings in BI-78D3. Loop-11, which represents the site of most significant divergence within the DRS (Fig. 3c), may also contribute to the discrimination exhibited by BI-78D3, a notion supported by the MD simulations that suggest a possible inter-action between the nitro group of BI-78D3 and T158, a residue unique to ERK1/2 in loop-11. As noted, the removal of this nitro group eliminates adduct formation.

We showed that an incubation of A375 cells with 1 µM BI-78D3 for 1 h, followed by buffer exchange and a 30 min stimu-lation of cells with EGF, results in the suppression of the

phosphorylation of ERK on the activation loop and p90RSK on S359/S364 and T573 (Fig. 4c)[57].

ATP-competitive inhibitors of ERK have been shown to induce G1 cell cycle arrest in a number of cancer cell lines, and at higher concentrations they induce apoptosis[23]. We discovered that a single dose of BI-78D3 (ED₅₀ ~ 6 µM, 1 h) induces a marked G1 arrest (48 h) and inhibits the proliferative potential of melanoma A375 cells, as measured using proliferation and colony formation assays over a period of days. Furthermore, after 72 h, measurable apoptosis is detectable in a significant fraction of cells, as deter-mined by caspase 3/7 activity and Annexin V (Fig. 4f, g). Notably, BRAF mutant cell lines that have acquired resistance to the BRAF inhibitor SB-590885 remain sensitive to BI-78D3. Significantly, the effect of BI-78D3 on colony formation is blunted by the overexpression of ERK2 C159A in both HEK and A375 cells, providing support for the idea that ERK is a target for BI-78D3 in these cells. This is further supported by the observation that cells expressing wild-type BRAF and KRAS, as well as mutant KRAS are significantly more resistant to BI-78D3 than mutant BRAF-expressing cells (Supplementary Fig. 24).

We showed that a daily intraperitoneal administration of BI-78D3 (15 mg kg⁻¹) strongly suppressed the growth of A375

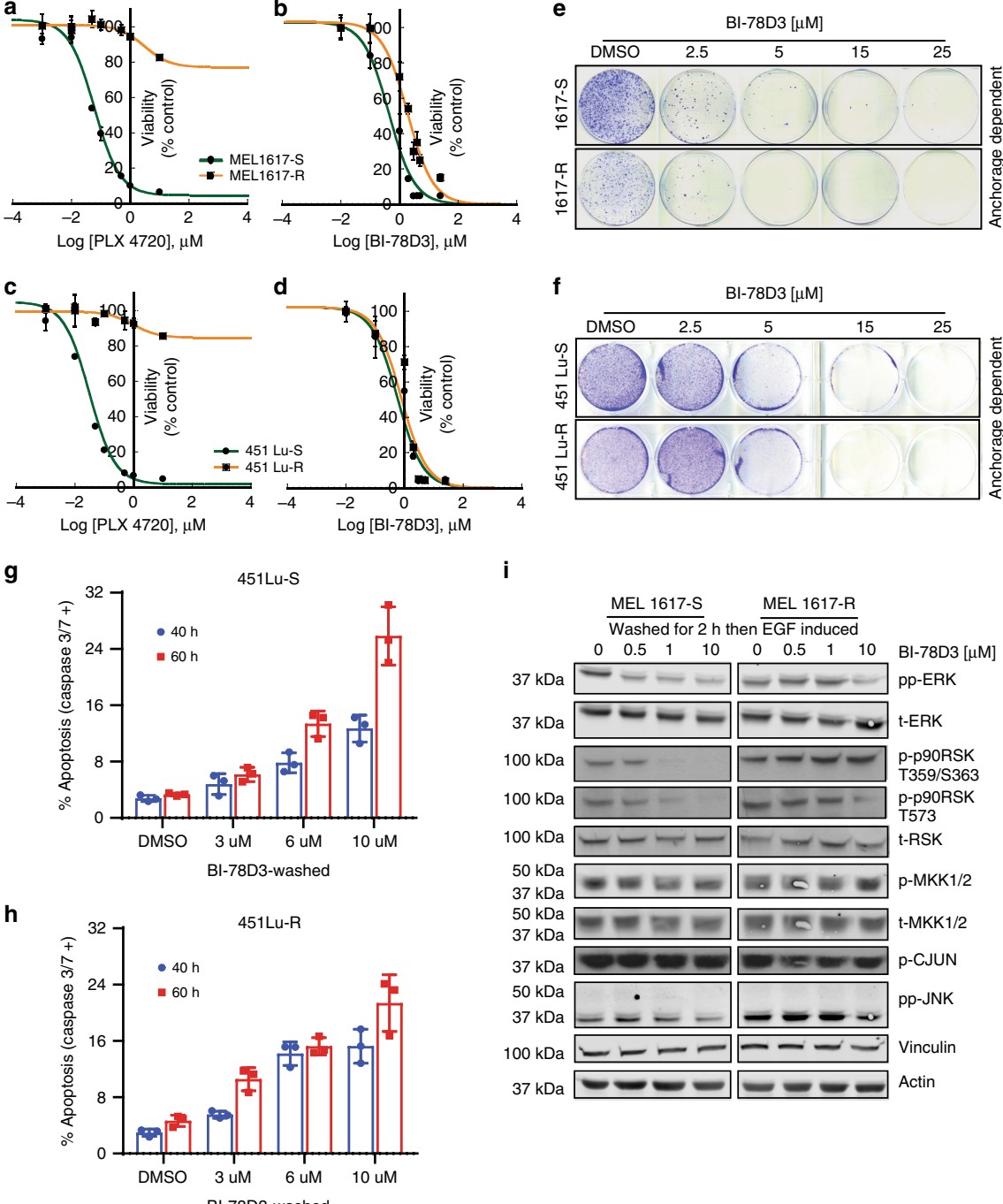

**Fig. 6** BI-78D3 overcomes acquired resistance to BRAF inhibition in melanoma cell lines. **a–d** BI-78D3 is equally potent in PLX4032-sensitive (-S) or -resistant (-R), *BRAF*-mutant melanoma cell lines. Cells were treated with either PLX4032 or BI-78D3 as shown and incubated for 72 h before analysis using an MTS assay (Error bars: ±SD ($n = 3$)). **e, f** BI-78D3 inhibits the anchorage-dependent growth of PLX4032-sensitive and -resistant melanoma cell lines (experiments were repeated at least two times). **g, h** BI-78D3 induces caspase-3/7 activity in 451 Lu-S and 451 Lu-R cell lines. Cells were analyzed using an IncuCyte caspase-3/7 reagent and the IncuCyte® ZOOM at the indicated time points. (IncuCyte imaging. Error bars: SEM ($n = 3$)). **i** Western blot analysis to evaluate the suppression of EGF-stimulated cell signaling in melanoma cell lines by BI-78D3. Cells were serum starved overnight then treated with BI-78D3 for 1 h in serum-free media, followed by inhibitor washout for 2 h in the same media. Cells were then stimulated with EGF for 30 min in full media. Images have been cropped for presentation; the uncropped images are shown in Supplementary Fig. 27

tumors in nude mice. Importantly we saw strong suppression of ERK but not JNK phosphorylation in the tumors 3 h after administration of BI-78D3, as judged by western blotting, supporting the notion that BI-78D3 exhibits selectively towards ERK and that the adduct is stable in vivo. The observed in vivo potency is consistent with the reported half-life of ERK1 and

ERK2 of 50 h[58], with the relative stability of the ERK adduct and with the in vitro cell experiments suggest that a single exposure of cells to BI-78D3 can suppress ERK phosphorylation for at least 8 h (Fig. 4b).

In summary, the covalent targeting of a cysteine in the D-recruitment site (DRS) of ERK1 and 2 represents a new strategy

for inhibiting their signaling in vivo. C159 (in ERK2), which lies within a critical interface recognized by many ERK binding partners, forms a covalent tetrahedral adduct with an amidine functional group of the small molecule BI-78D3 in vivo. The adduct is relatively stable to reversible cysteine elimination and exhibits an ability to impede both the activation of ERK, as well as its downstream signaling for several hours in mammalian cells. Steric constraints appear to confer selectivity on the reaction of BI-78D3 with ERK, as no other MAPK tested, that contain this conserved cysteine, is covalently modified by BI-78D3 in vitro. We show that BI-78D3 has antitumor activity in BRAF/MEK inhibitor-naïve and BRAF/MEK inhibitor-resistant melanoma cells containing a BRAF mutation, demonstrating its obvious potential to overcome acquired resistance to BRAF and MEK inhibitors. Transient overexpression of the C159 to Ala mutant of ERK2 confers resistance of HEK293 and A375 cells to BI-78D3, further validating the DRS of ERK1/2 as the target of BI-78D3 in cells. The DRS of ERK is an attractive site to target with small molecules because it offers the possibility of countering the effect of suppressing feedback inhibition by ERK (through inhibiting the action of MEK on ERK), a source of resistance to inhibitors of MEK and BRAF and would be expected to exhibit different resistance profiles. Further mechanistic studies are warranted to provide a roadmap for fine-tuning the chemical properties of BI-78D3 to improve its specificity and potency in vivo and to explore the potential for developing new amidine functionalities to modify other proteins of interest covalently.

## Methods

All buffers and solutions used in this study were degassed by ARGON (Ar) gas (AirGas) for at least 10 min.

**Synthesis of (Con-1) or (4).** The synthetic procedures of Methyl 2-((4-(2,3-dihydrobenzo[b][1,4]dioxin-6-yl)-5-oxo-4,5-dihydro-1H-1,2,4-triazol-3-yl)thio)thiazole-5-carboxylate (Con-1) (4), NMR spectra and high-resolution mass spectrometry spectrum are provided in the supplementary methods section and Supplementary Figs. 30–32.

**NMR assignments of purchased reagents.** BI-78D3 (1) (Chemdea, Cat # CD0179 or Cayman Chemical, Cat # 21183) $^1$H NMR (600 MHz, phosphate buffer with 50% DMSO-$d_6$): $\delta$ 8.32 (s, 1H), 6.80 (d, $J$ = 8.3 Hz, 1H), 6.60 (m, 2H), 4.12 (m, 4H). $^{13}$C NMR (125 MHz, phosphate buffer with 50% Dioxane-$d_8$, Bruker Advance III): $\delta$ 169.48, 157.97, 148.56, 144.28, 143.96, 143.73, 136.34, 125.4, 120.58, 117.64, 116.25, 64.41, 64.34.

2-Methoxyethanethiol (2) (Enamine, Cat # EN300–35885) Major component $^1$H NMR (600 MHz, phosphate buffer with 50% DMSO-$d_6$): $\delta$ 3.4 (t, 2H), 3.18 (s, 3H), 2.54 (t, 2H).

1-Hydroxyethanethiol (SigmaAldrich, Cat # M3148) $^1$H NMR (600 MHz, phosphate buffer with 50% DMSO-$d_6$): $\delta$ 3.49 (t, 2H), 2.47 (t, 2H).

Ethanethiol (Acros Organics, Cat # 117865000) $^1$H NMR (600 MHz, phosphate buffer with 50% DMSO-$d_6$): $\delta$ 2.64 (q, 2H), 1.33 (t, 3H).

**NMR assignments of adduct (3).** Adduct 3 is formed between BI-78D3 and 2-methoxyethanethiol (Supplementary Fig. 9). Major component $^1$H NMR (600 MHz, phosphate buffer with 50% Dioxane-$d_8$): $\delta$ 8.44 (s, 1H), 6.85 (d, $J$ = 8.6 Hz, 1H), 6.73 (d, $J$ = 2.3 Hz, 1H), 6.68 (dd, $J$ = 2.3, 8.6 Hz, 1H), 4.22 (no resolution from the water peak), 3.67 (t, 2H), 3.45 (t, 2H), 3.27 (s, 3H). $^{13}$C NMR (125 MHz, phosphate buffer with 50% dioxane-$d_8$, Bruker Advance III): $\delta$ 175.68, 159.13, 156.59, 147.25, 143.98, 143.32, 143.14, 127.67, 121.7, 117.47, 117.04, 69.7, 64.4, 64.3, 58.1, 34.4.

**Protein expression and purification.** Constitutively active MKK1, full-length ERK2 (Rattus norvegicus mitogen-activated protein kinase 1, GenBank accession number NM_053842) or different ERK2 mutants were purified from bacteria and where applicable, activated following our published protocol[59]. Full-length ERK1 was expressed, purified, and activated as described in Callaway et al.[57]. Full-length human JNK1α1 (GenBank accession number NM_002750), human JNK2α2 (GenBank accession number NM_002752), or N-terminal truncated human JNK3α2 (amino acids 39–422 with alanine inserted between amino acids 39 and 40, GenBank accession number NM_138982) were expressed, purified and activated as we described somewhere else[60,61]. p38MAPKα was expressed, purified and activated as described previously[62]. Human ERK5 (full length) was purchased from

SignalChem, and GST-ERK5 kinase domain (aa 31–391, GenBank accession number NP_002740.2) was expressed and purified following GE Healthcare protocol for GST-tagged proteins purification. All the kinases were prepared in buffers free from any reducing agent and stored in buffer S [25 mM HEPES (pH 7.5), 50 mM KCl, 0.1 mM EDTA, 0.1 mM EGTA, and 10–20% (v/v) glycerol] at −80 °C.

GST-ATF2 (1–115), GST-c-Jun (1–221) and His-Ets-1 (1–138) were expressed and purified following our published protocols[63]. All were prepared in buffers free from any reducing agent and stored in buffer S.

**Kinase activity assay.** Assays were conducted in kinase assay buffer (25 mM HEPES buffer-pH 7.5, 50 mM KCl, 0.1 mM EDTA, 0.1 mM EGTA and 5% DMSO), containing 10 μg ml$^{-1}$ BSA, 10 mM MgCl$_2$ and different concentrations of BI-78D3. Each reaction was started by the addition of 500 μM radiolabeled [γ-$^{32}$P] Mg$^{2+}$-ATP (100–1000 c.p.m. pmol$^{-1}$). For ERK assays, 4 nM WT-ERK1/2 or mutant ERK2 was assayed with 10 μM Ets1 (1–138), or 10 μM D-sub peptide, FQRKTLQRRNLKGLNLNL-XXX-TGPLSPGPF (X is 6-aminohexanoic acid)[42]. For JNK assays, 25 nM active JNK1α1, JNK2α2, or JNK3α2 were assayed with 2 μM GST-c-Jun (1–221) or 10 μM GST-ATF2 (1–115) protein substrates. For the MKK1 assays, 100 nM constitutively active MKK1 was assayed with 1000 nM full-length inactive ERK1 or ERK2 as substrate, assays were conducted in the same kinase assay buffer except each reaction was started by the addition of 1 mM regular ATP. Kinase activity was quantified at different inhibitor concentrations by determination of the initial rates of the reaction. For all assays except MKK1, at each time point (0.5, 1, 1.5, 2, 4 min) 10 μl aliquots were withdrawn from every reaction and spotted on 2 × 2 cm$^2$ squares of P81 cellulose paper; the papers were washed for 3 × 15 min in 50 mM phosphoric acid (H$_3$PO$_4$) then the c.p.m. associated with each paper were quantified in a PerkinElmer Tri-Carb Liquid Scintillation Counter. For MKK1 assays, at each time point (0.5, 1, 2, 4, 10 min), 25 μl aliquots were withdrawn and quenched in boiling 5× PAGE loading dye for 10 min. The quenched samples were fractionated on a 10% SDS-PAGE gel, before western blotting, in order to quantify the amount of double-phosphorylated ERK in each sample using a Licor detection system.

For the dialysis experiment, 5 μM of each activated enzyme was treated with 100 μM BI-78D3 or DMSO for 1 h in the kinase assay buffer. Reaction mixtures were dialyzed overnight in the kinase assay buffer without DMSO (4000 folds), the protein concentrations were determined using the Bradford reagent (Bio-Rad). The activities of the enzymes towards their substrates were estimated after and before the dialysis process, following the protocol that is mentioned above.

**Fluorescence anisotropy displacement assay.** Assays were performed in 25 mM HEPES (pH 7.5), 50 mM KCl, 0.1 mM EDTA, 0.1 mM EGTA, 40 μg ml$^{-1}$ BSA containing 40 nM FITC-Ste7 peptide, 2 μM activated ERK2-WT or C159S mutant, and different concentrations of BI-78D3 in a final volume of 100 μl. After equilibration at 27 °C, samples were excited with both horizontally and vertically polarized light at 490 nm, and the emission recorded every 5 s for a total of 100 s at 517 nm. The slit width of the excitation and emission were adjusted to 1 nm with a 350 ms integration time. The average anisotropy values were calculated and fit to previously reported equations[64] using GraphPad Prism 7.04.

**Time-dependent inhibition.** Different concentrations of BI-78D3 (0, 0.5, 1, 3, 12, 80, and 200 μM) were preincubated at room temperature with 200 nM active ERK2 for different time points (0, 0.5, 5, 15, 30 and 60 min) in kinase assay buffer (25 mM HEPES buffer-pH 7.5, 50 mM KCl, 0.1 mM EDTA, 0.1 mM EGTA). At each time point, aliquots were diluted 200-fold into another tube containing 20 μM Ets-1, 500 μM [γ-32P] ATP (100–1000 c.p.m. pmol-1) and 11 mM MgCl$_2$ (final concentrations) and the activity of ERK was determined. The percentage of the remaining activity was plotted against the preincubation time at each inhibitor concentration. $K_i$ and $k_{inact}$ values were determined using nonlinear regression of the negative slopes of the inactivation plots (the natural logarithm of the residual activity versus preincubation time) versus inhibitor concentrations using KaleidaGraph software and Eq. (1)[65].

$$k_{obs}^{inact} = \frac{k_{inact}[I]}{K_i + [I]}, \qquad (1)$$

**Protein NMR.** Uniformly $^2$H, $^{15}$N, ILV-labeled inactive ERK2 was expressed and purified as described previously[39]. All the NMR experiments were performed in a buffer containing 50 mM phosphate, 150 mM NaCl and 200 μM EDTA at pH 6.8 (10% D$_2$O). The DTT present in the early stages of the purification process was eliminated using multiple dilution steps using spin columns with a 10 kDa cutoff. Chemical shift perturbations were evaluated from 2D $^{15}$N, $^1$H TROSY spectrum acquired at 800 MHz on a 116 μl sample containing 0.4% of $d_6$-DMSO, with and without 130 μM ligand utilizing previously determined backbone assignments of ERK2[40,66]. The same samples were used to determine methyl perturbations utilizing $^{13}$C, $^1$H SOFAST-HMQC experiments also at 800 MHz. Methyl assignments were transferred from Xiao et al.[41], and chemical shift perturbations were

calculated using Eq. (2)[39].

$$\Delta\delta_j = \sqrt{0.5 \sum_{i=1}^{2} \left(\frac{\Delta\delta_{ji}}{\sigma_{ik}}\right)^2},\qquad(2)$$

where the index $j$ identifies a particular residue, $i$ identifies a particular nucleus ($^1$H, $^{15}$N or methyl $^{13}$C), and $\sigma_{ik}$ is the standard deviation for atom type $i$ and amino acid type $k$ obtained from the Biological Magnetic Resonance Databank (BMRB).

The effect of the ligand on the resonance corresponding to C159 was investigated by titrating a 200 μM sample of $^2$H, $^{15}$N-labeled ERK2 with increasing amounts of ligand to a final concentration of 0 (a spectrum of free ERK2 containing 2.5% $d_6$-DMSO was used as reference), 49, 97, 145, 192, 330 μM, introduced as 5 μl aliquots from 3 and 9 mM stock solutions in $d_6$-DMSO. The effect of DMSO alone (final concentration: ~7.7%) was evaluated by the addition of DTT to a final concentration of 10 mM at the end of the titration series. DTT abolishes the ERK2/BI-78D3 interaction. 2D $^{15}$N, $^1$H TROSY experiments were collected for each titration point at 600 MHz.

All NMR experiments were collected at 25 °C on Bruker Avance III HD and Varian Inova instruments equipped with cryogenic probes capable of applying pulsed field gradients on the z-axis. Data were processed using NMRPipe[67] and analyzed using Sparky[68].

**Ellman's reagent assays.** Ellman's reagent was employed to examine the effect of BI-78D3 on protecting different ERK2 sulfhydryl groups. Cysless ERK2 or cysless ERK2 bearing only one or two cysteines at C159 and/or C164 was incubated with BI-78D3 for 60 min before exposure to Ellman's reagent, following the manufacturer's protocol (Thermo Scientific-Cat number 22582). 5 μM recombinant ERK2 was incubated with 100 μM BI-78D3, or DMSO for 60 min, excess compound was washed with a PD 10 desalting column (GE Healthcare), and protein was concentrated to 50 μM. Two hundred microliters of the concentrated protein sample in 100 mM sodium phosphate buffer (pH 8.0) containing 1 mM EDTA was transferred to a cuvette containing 4 μl of 10 mM Ellman's reagent and absorbance at 412 nm were measured using Varian's Cary 50 spectrophotometer (Agilent Technology) after 15 min of incubation at room temperature. The reported molar absorptivity (molar extinction coefficient) of (2-nitro-5-thiobenzoic acid), the product of the reaction of Ellman's reagent and any free sulfhydryl group at 412 nm is 14,150 M$^{-1}$ cm$^{-1}$.

To estimate the number of surface cysteines in different proteins, His tagged ERK2, JNK1, JNK2, JNK3, JNK2 C163A mutant, p38-α MAPK, GST-ERK5 (31–391) kinase domain, and Aldolase A were expressed and purified as described before in the Methods section. Storage buffer was exchanged by excessive dialysis (10× of 1/4000 dilution) in 50 mM phosphate buffer (pH 7.5) containing 10% glycerol (p38-α MAPK and GST-ERK5 were dialyzed in the same buffer without glycerol). BSA was purchased from Fisher Bio-reagent, Esterase was purchased from Sigma Aldrich, and Lysozyme was purchased from MB Biomedicals. The three proteins were re-solubilized in 50 mM phosphate buffer (pH 7.5), the buffer was further exchanged by excessive dialysis (10× of 1/4000 dilution) in 50 mM phosphate buffer (pH 7.5). After dialysis, each protein was concentrated to 60 μM at least. One hundred microliters containing 50 μM of each protein sample and 1 mM EDTA and 2 μl 10 mM Ellman's reagent were added in a 100 μl quartz cuvettes, and the absorbance at 412 nm was recorded using Varian's Cary 50 spectrophotometer (Agilent Technology) after 15 min of incubation at room temperature. The recorded absorbance at A$_{412}$ nm was applied to a calibration curve that constructed using different concentrations of L- Cysteine hydrochloride (Sigma-Aldrich) to quantify the number of free cysteines in each tested protein.

To assess the stability of the ERK·BI-78D3 adduct, 5 μM recombinant cysless ERK2 bearing only C159 was incubated with DMSO (control sample) or 100 μM BI-78D3 for 60 min in 100 mM sodium phosphate, pH 8 containing 1 mM EDTA. The reaction mixture was desalted using a PD 10 column (GE Healthcare), and labeled protein was concentrated to 50 μM in the same buffer. The resulting adduct was incubated at room temperature, and at given time points a fraction of the adduct was desalted using a Bio-spin column (Bio-Rad), concentrated to 50 μM and 200 μl was transferred to a cuvette containing 4 μl of 10 mM Ellman's reagent and the absorbance at 412 nM recorded after 15 min.

**UV-visible spectrophotometry.** UV-Vis spectra of the reaction of 100 μM of each thiol or 50 μM of ERK2 and 10 μM BI-78D3 were obtained at 24 °C on an Agilent 8453 diode-array spectrophotometer, in 50 mM phosphate buffer, pH 7.5 and 2% Dioxane. The reaction was started by the addition of BI-78D3 and spectra were acquired for 1000 s (every 10 s).

For the dialysis experiment, the storage buffer of each protein was exchanged by excessive dialysis (10× of 1/4000 dilution) in 50 mM phosphate buffer (pH 7.5) containing 10% glycerol, 5 μM of each enzyme (ERK2, JNK1, JNK2, JNK3, JNK2 C164A, and Aldolase A) were incubated with DMSO or 100 μM BI78D3 for 30 min in 2% Dioxane/phosphate buffer pH 7.5, 10% glycerol, then dialyzed two times overnight (1/4000 dilution each) in 50 mM phosphate buffer (pH 7.5) containing 10% glycerol. Samples were concentrated to 50 μM before recording the UV spectra.

**Intact protein analysis.** In each experiment, 5 μM WT-ERK2, ERK2 mutants, WT-JNK1/2, WT-p38MAPKα or ERK5 catalytic domain protein were incubated ±100 μM BI-78D3 in 25 mM HEPES buffer (pH 7.5) containing 50 mM KCl, 0.1 mM EDTA, 0.1 mM EGTA and 5% DMSO for 20–120 min or overnight at room temperature. Then the reaction mixture was desalted, and the buffer was exchanged to 25 mM ammonium bicarbonate buffer. Samples were flash frozen and stored at −80 °C for later analysis, or instantly injected in the liquid chromatography-mass spectrometry (LC-MS) instrument. 100–200 pmol of WT-ERK2, Flag-ERK2, p38MAPKα or ERK5 ±BI-78D3 were analyzed in the (LC-MS) on an Orbitrap Fusion mass spectrometer (Thermo-Fisher). Following the acquisition, data were deconvoluted using MagTran and Protein Deconvolution 4.0. In some other cases, 100–200 pmol of WT-ERK2, ERK2 mutants or WT-JNK1/2 ±BI-78D3 were analyzed in 4000 Q TRAP mass spectrometer (Applied Biosystems, Foster City, CA) coupled with an online high-performance liquid chromatography system (Shimadzu, Columbia, MD).

**Cell culture.** Human Embryonic Kidney cell line (HEK-293T, ATCC CRL-3216) and melanoma cell line (A375, ATCC CRL-1619) were purchased from the American type culture collection (ATCC; Manassas, VA). 451 Lu and MEL 1617 melanoma cell lines were provided by Prof. Kenneth Tsai, H. Lee Moffitt Cancer Center & Research Institute, Florida, USA. HEK293T cells were cultured in DMEM medium (Life Technologies) supplemented with FBS (FBS; 10%; US-Origin, heat inactivated, Life Technologies), while melanoma cell lines were cultured in RPMI-1640 medium (Life Technologies) supplemented with FBS (FBS; 5 or 10%; US-Origin, heat inactivated, Life Technologies), 100 U ml$^{-1}$ penicillin and 100 μg ml$^{-1}$ streptomycin (Life Technologies) and maintained in a 37 °C incubator with 5% CO$_2$. RAF-inhibitor-resistant cell lines were maintained in the same media containing 1 μM of PLX4720 (Selleckchem, Cat # S1152). For the specificity study, human pancreas ductal adenocarcinoma cell line (MIA PaCa-2, ATCC CRL-1420), human breast adenocarcinoma cell line (MCF7, ATCC HTB-22) and human non-small cell lung cancer cell line (A549, ATCC CCL-185) were purchased from the American type culture collection (ATCC; Manassas, VA). Human Glioma cell line (U87-MG, ATCC HTB-14) was obtained from the Neurosurgery Tissue Bank, University of California, San Francisco, USA. Human breast adenocarcinoma cell line (MDA-MB-468, ATCC HTB-132) was provided as a gift from Dr. Chandra Bartholomeusz, The University of Texas MD Anderson Cancer Center, Houston, TX, USA. MIA PaCa-2 and U87-MG cell lines were cultured in DMEM medium (Life Technologies) while the human non-small cell lung cancer cell lines were cultured in RPMI-1640 medium and the human breast adenocarcinoma cell lines were cultured in DMEM/F12 medium. All media were supplemented with FBS (FBS; 10%; US-Origin, heat inactivated, Life Technologies), 100 U ml$^{-1}$ penicillin and 100 μg ml$^{-1}$ streptomycin and maintained in a 37 °C incubator with 5% CO$_2$. Cells that were purchased from ATCC were authenticated by the ATCC using short, tandem-repeat profiling. Cells that were provided by research groups in The University of Texas MD Anderson Cancer Center were validated by STR DNA fingerprinting using the AmpFℓSTR Identifiler kit according to the manufacturer's instructions (Applied Biosystems, Grand Island, NY). The rest of the cell lines were verified using Human 9-Marker STR Profile and Interspecies Contamination Test—IDEXX BioResearch. Mycoplasma tests were performed monthly using MycoAlert™ Mycoplasma Detection Kit (Lonza).

**Transfection and stable cell lines.** For the transient transfection, HEK293T and A375 cell lines were transfected with pcDNA3 vector or the pcDNA3 vector containing DNA encoding for WT-ERK, ERK2 (C159A), or ERK2 (C164A) by lipofectamine 3000 (Invitrogen, California, USA), according to the manufacturer's protocol. After 48 h of transfection, cells were treated as specified in each experiment. For the generation of stable cell lines, ERK2 cDNAs were shuttled into pLenti-C-Myc-DDK-IRES-Puro (OriGene, Rockville, MD, USA) plasmid following the OriGene Precision Shuttling protocol after digestion with SgfI and MluI. Cells overexpressing ERK2 were generated by lentiviral infection with the Lenti-vpak Packaging Kit (OriGene), following the manufacturer's instructions. Control cells were prepared using the same method, but with an empty control plasmid. Cells were selected with 2 μg ml$^{-1}$ puromycin to establish stable cell lines.

**In cell labeling of ERK2.** To analyze the mass shift of ERK2 in HEK293 cells following treatment with BI-78D3, HEK293 cells stably expressing WT Flag-ERK2 were serum starved overnight and treated with DMSO or 25 μM BI-78D3 for 2 h in serum-free media. Cells were lysed in lysis buffer (50 mM Tris HCl, pH 7.4, with 150 mM NaCl, 1 mM EDTA, and 1% TRITON X-100) containing a protease and phosphatase inhibitor cocktail (Thermo Scientific). The lysates were centrifuged at 14,000 rpm at 4 °C for 15 min. The supernatant was collected and incubated immediately with anti-Flag M2 affinity Gel (Sigma-Aldrich) for 1 h following the manufacturer's protocol. Flag-ERK2 was eluted with 3× flag peptide (Sigma-Aldrich), buffer was exchanged to 25 mM ammonium bicarbonate using a desalting column, then protein was concentrated and frozen instantly until its intact masses verified using ESI-Mass spectrometry.

**Cell viability assays.** Cell viability was measured by the CellTiter 96® AQueous One Solution Reagent (Promega) containing [3-(4,5-dimethylthiazol-2-yl)-5-(3-

carboxymethoxyphenyl)-2-(4-sulfophenyl)-2H-tetrazolium, inner salt; MTS] and an electron coupling reagent (phenazine ethosulfate; PES). Briefly, 2000 cells per well in 96-well plates were incubated for 24 hours to adhere and then titrated with varying concentrations of BI-78D3 or PLX4720 (Selleckchem, Cat# S1152) in full media without antibiotics in a final volume of 100 μl media. At indicated time points, each well was treated with 20 μl of the reagent, incubated for an additional 1–4 h before measuring the absorbance at 490 nm using the Victor³V Model 1420 plate reader from PerkinElmer (Waltham, MA). The measured absorbance is directly proportional to the number of living cells in culture.

**IncuCyte proliferation assay.** After seeding 2000 cells in each well of a 96-well plate, cells were maintained in a 37 °C incubator with 5% $CO_2$ for 18–24 h to adhere. Cells were treated with different doses of BI-78D3 in full media containing 0.1% DMSO (100 μl/well) without antibiotics. When washout as indicated, media containing BI-78D3 were changed to a full fresh media 1 h after treatment. Cells were incubated and analyzed using Incucyte ZOOM microscope (image recorded every 1 h with a ×10 objective) maintained in a 37 °C incubator with 5% $CO_2$. The real-time growth confluence was estimated and plotted using GraphPad Prism software.

**IncuCyte single-cell apoptosis analysis.** Similar to the proliferation assay, 2000 cells were seeded in each well of a 96-well plate, then incubated for 24 h to adhere. Cells were treated with different doses of BI-78D3 in antibiotic-free full media containing 0.1% DMSO and constant concentration of the lncuCyte Caspase-3/7 or Annexin V reagent following the manufacturer's protocol. When washout was indicated, cells were treated initially with BI-78D3 for 1 h; then the media has been changed with new full media containing the lncuCyte Caspase-3/7 or Annexin V reagent. Cells were incubated and imaged in the lncuCyte® ZOOM equipment with a ×10 objective at indicated time points. The number of apoptotic cells was normalized to the percentage confluency at the final time point to account for cell proliferation.

**Soft agar assay.** Anchorage-independent growth was determined as described previously[69]. Briefly, A375 or HEK293T cells were serum starved overnight, then treated with different doses of BI-78D3 for 2 h in serum-free media before trypsinization. For the A375 cell line, 2500 cells were re-suspended in 0.35% SeaPlague GTG agarose growth medium (RPMI media with 5% FBS) and then plated in six-well plates containing solidified 0.7% SeaPlague GTG agarose in growth medium (RPMI media with 5% FBS). For the HEK293T cell line, 10,000 cells were resuspended in 0.3% SeaPlague GTG agarose growth medium (DMEM media with 10% FBS) and then plated in 60 mm TC plates containing solidified 0.7% SeaPlague GTG agarose in growth medium (DMEM media with 10% FBS).

**Colony formation assay.** A375, HEK293T, MEL-1617, and 451 Lu cells were serum starved overnight then treated with different doses of BI-78D3 for 2 h in serum-free media before trypsinization. Cells were re-plated (2500 cells per well) in six-well plate containing RPMI media with 5% FBS for A375 or 10% FBS for MEL-1617 and 451 Lu cell lines. HEK293T cells were re-plated (10,000 cells per plate) in 60 mm TC plates in DMEM media with 10% FBS. Cells were allowed to grow for 14 days. Cells were then fixed for 15 min in 4% paraformaldehyde and stained with 0.2% crystal violet (in 20% methanol) for 10 min. Cells were washed several times with distilled water, and colonies formed were counted.

**Cell cycle analysis.** A375 cells were treated with DMSO or 6 μM BI-78D3 for 24, 48 or 72 h in full media without antibiotics. For the washout experiments, cells were washed by 1× warm PBS buffer after 1 h of DMSO or BI-78D3 treatment, then incubated in full fresh media for the rest of the experiment. Cells were harvested and washed twice with phosphate-buffered saline (PBS) and fixed in ice-cold 70% ethanol overnight at −20 °C. The next day cells were washed 2× with PBS and resuspended in propidium podide (PI)/RNase staining solution (Cell Signaling Cat # 4087) for 30 min at room temperature and protected from light. Cell-cycle analysis was performed using a BD LSRFortessa SORP Flow Cytometer. Data obtained from the cell cycle distributions were analyzed using both FlowJo v10 (Tree Star, Inc., Ashland, OR, USA) software and ModFit 3.0 (Verity) software. FlowJo v10 was employed to estimate the percentage of cells in G1, S, and G2. ModFit 3.0 software was used to determine the percentage of cells in apoptosis (the sub-G1) population.

**Immunofluorescence study.** A375 cells were maintained in RPMI media (Invitrogen) supplemented with 5% (v/v) FBS (Invitrogen), 100 U ml⁻¹ penicillin (Invitrogen), and 100 g ml⁻¹ streptomycin (Invitrogen). Cells were cultured in a humidified 5% $CO_2$ incubator at 37 °C. 25,000 cells were placed on poly-L-lysine-coated coverslips seeded in 24-well plates and incubated to adhere for 24 h. pcDNA-ERK2-WT or mutant was transfected using Lipofectamine™ 3000 (Invitrogen) reagent according to the manufacturer's protocol. Cells were incubated for another 36 h, then maintained for one more night in serum-free media. Cells were treated with 25 μM of BI-78D3 for 2 h and induced with EGF for 15 min, followed by washing using PBS and fixation in 4% paraformaldehyde in PBS for 10 min at

room temperature. The cells were permeabilized with 0.2% Triton X-100 for 5 min and blocked with 10% Goat serum (Cell Signaling Technology). Image-iT™ FX signal enhancer (Invitrogen) was used according to the manufacturer's protocol. Immunostaining was performed by incubating cells with monoclonal ANTI-FLAG® M2-FITC antibody produced in mouse (F4049-Sigma). DAPI (4′, 6′-dia-midino-2-phenylindole) was included in the mounting medium as a counterstain for nuclei and was visualized by the A Zeiss Axiovert 200M microscope with ×63 magnification.

**Animal studies.** Mice were handled in accordance with a protocol approved by the Institutional Animal Care and Use Committee (IACUC) of the University of Texas at Austin. All mice were allowed to acclimate for at least 1 week before use in experiments. Inbred, athymic nude mice were injected with $1 \times 10^6$ A375 malignant melanoma cells into the right flank. Tumors were palpable 3 days after injection. When tumors reached an average volume of 150 mm³, mice were randomized into two groups of ten mice each. One group was treated with freshly prepared vehicle (2.5%EtoH, 5%Tween-80, 1×PBS), and one group was treated with 15 mg kg⁻¹ BI-78D3 dissolved in the same vehicle. Mice were treated daily by intraperitoneal injection with daily measurements for tumor volume. Mice were treated until tumors reached a maximum diameter of 1.5 cm as prespecified by the University of Texas IACUC-approved protocol. Tumor volume was estimated by the formula: Estimated tumor volume (mm³) = $0.5236 \times L1 \times (L2)^2$, where L1 and L2 are the long and short diameter, respectively. Five mice from each group were euthanized to obtain tumor tissue for molecular analysis by western blotting and the peptide-based biosensor[53]. The tumor measurements of these mice were analyzed for tumor growth comparison.

**Immunohistochemistry staining.** FFPE sections were stained with an antibody against pp-ERK1/2 (Thr202/Tyr204—clone 20G11/Cell Signaling) by the MD Anderson Cancer Center Research Histology Core Lab (NCI CA#16672). Areas of tumor sections were outlined in a blinded fashion (K.Y.T.) and staining quantified as percent tumor cells positive for staining using ScanScope on the Aperio platform.

**Western blotting.** For western blotting, cells were seeded and incubated to adhere for 24 h. Cells were exposed to serum-free media for the indicated times, then treated with DMSO or different doses of BI-78D3 in serum-free media. Cells were induced with 100 nM EGF (Invitrogen) for 15–30 min before lysis. For the washout experiments, cells were washed by 1× warm PBS buffer after 1–2 h of DMSO or BI-78D3 treatment, then maintained in media without antibiotics for the indicated time before induction by 100 nM EGF (Invitrogen) for an extra 15–30 min. To lyse, the cells were washed by 1× ice-cold PBS (Invitrogen) then lysates were prepared in M-PER ™ Mammalian Protein Extraction Reagent (Thermo-Fisher) containing 1× protease and phosphatase inhibitors cocktail (Thermo-Fisher). The lysates were centrifuged at 13,000 rpm for 20 min, and the total protein content in each supernatant was estimated by Bradford analysis (Bio-Rad). 50–80 μg of each lysate was fractionated on a 10% SDS polyacrylamide gel (Bio-Rad) and transferred to Hybond-P PVDF membrane (GE Healthcare) or Immobilon-FL membrane (Millipore). Primary antibodies were incubated overnight at 4 °C using 1:2000 anti-phospho-p44/42 MAPK (ERK1/2) (Thr202/Tyr204) (E10) mouse mAb (Cat # 9106, Cell Signaling Technology); 1:1000 anti p44/42 MAPK (ERK1/2) (137F5) rabbit mAb (Cat # 4695, Cell Signaling Technology); 1:1000 anti-DYKDDDDK Tag (9A3) mouse mAb (Cat # 8146, Cell Signaling Technology); 1:1000 anti-phospho-p90RSK (Thr573) rabbit polyclonal Abs (Cat # 9346, Cell Signaling Technology); 1:1000 anti-phospho-p90RSK (Thr359/Thr363) rabbit polyclonal Abs (Cat # 9344, Cell Signaling Technology); 1:1000 anti-RSK1/RSK2/RSK3 (32D7) Rabbit mAb (Cat # 9355, Cell Signaling Technology); 1:500 anti-phospho-Elk-1 (Ser-383) rabbit polyclonal IgG (Invitrogen); 1:1000 anti-phospho-MEK1/2 (Ser217/221) rabbit mAb (Cat # 9154, Cell Signaling Technology); 1:1000 anti-MEK1/2 (L38C12) mouse mAb (Cat # 4694, Cell Signaling Technology); 1:2000 anti-phospho-SAPK/JNK (Thr183/Tyr185) (G9) mouse mAb (Cat # 9255, Cell Signaling Technology); 1:2000 anti-JNK2 (56G8) rabbit mAb (Cat # 9258, Cell Signaling Technology); 1:10,000 anti-phospho-c-Jun (Ser-63), clone Y172 rabbit mAb (Cat # 04–212, Millipore) or 1:1000 anti-phospho-c-Jun (Ser-63) II rabbit polyclonal Ab (Cat # 9261, Cell Signaling Technology); 1:1000 anti-c-Jun (60A8) rabbit mAb (Cat # 9165, Cell Signaling Technology); 1:10,000 anti-phospho-p38α (Thr180/Tyr182), clone 8.78.8 rabbit mAb (Cat # 05-1059, Millipore) or 1:1000 anti-phospho-p38 MAPK (Thr180/Tyr182) (D3F9) XP rabbit mAb (Cat # 4511, Cell Signaling Technology); 1:1000 anti-p38 MAPK (D13E1) XP rabbit mAb (Cat # 8690, Cell Signaling Technology); 1:1000 anti-phospho-BMK1/Erk5 (Thr218/Tyr220) rabbit polyclonal Ab (Cat # 07-507, Millipore); 1:1000 anti-Erk5 rabbit polyclonal Ab (Cat # 3372, Cell Signaling Technology); 1/2000 anti-Vinculin (E1E9V) XP rabbit mAb (Cat # 13901, Cell Signaling Technology) and 1:5000 anti-actin, clone 4 mouse mAb (Cat # MAB1501R, Millipore). Secondary anti-rabbit (Bio-Rad) or anti-mouse (Bio-Rad) horseradish peroxidase-conjugated secondary antibodies and Western Bright ECL Western Blotting Reagents (Advansta) were used to develop the blots. In certain experiments, fluorescent western staining was developed on the Odyssey fluorescent western system (Li-Cor); in these cases, secondary antibodies were from Li-Cor: IRDye 680RD goat anti-rabbit IgG and

IRDye 800CW goat anti-mouse IgG. Precision Plus Protein™ Dual Color marker (Cat #1610374, Bio-Rad) was used in all the experiments. Information about the replicates of each experiment is mentioned in detail in the figure legends.

**Fluorescence assays for A375 cell lysates**. The ERK selective sensor (named ERK-sensor-D1 or Sox-Sub-D), that we reported somewhere else[51], was employed to perform fluorescence kinase assay for ERK activity in the cell lysate. The assay was performed at room temperature in 96-well plate (60 µl reaction volume). The assay buffer containing 25 mM HEPES, pH 7.5, 50 mM KCl, 2 mM DTT, 0.1 mM EDTA, 0.1 mM EGTA, 10 mM MgCl2 was mixed with 20 µg protein of cell lysate prepared following the same protocol of lysate preparation for western blotting and 0.5 mM ATP. The reaction was started by addition of 20 µM ERK-sensor-D1 (Sox-Sub-D) and the fluorescence emission (Ex: 365, Em: 485) was recorded each 10 −15 s for 30 min using the Victor$^3$V model 1420 plate reader from PerkinElmer (Waltham, MA). The rate of the reaction was estimated using the slope of the product formation.

**Molecular dynamics simulations**. The detailed methods of molecular dynamic simulations are provided in the supplementary methods section.

**Reporting summary**. Further information on research design is available in the Nature Research Reporting Summary linked to this article.

## Data availability
Data supporting the findings of this study are available within the paper and its Supplementary Information files or available from the authors upon reasonable request. A reporting summary for this article is available as a Supplementary Information file.

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

## Acknowledgements

Financial support was from grants from the Cancer Prevention and Research Institute of Texas grant (RP160657 and RP180880), the Welch Foundation (F-1390) to K.N.D., the National Institutes of Health (R01 GM123252) to K.N.D. and R.G. and the Welch Foundation (F-1691) to P.R. The content is solely the responsibility of the authors and does not necessarily represent the official views of the National Institute of General Medical Sciences or the National Institutes of Health. Molecular graphics and analyses were performed with the UCSF Chimera package. Chimera is developed by the Resource for Biocomputing, Visualization, and Informatics at the University of California, San Francisco (supported by NIGMS P41-GM103311). The authors would like to acknowledge NIH grant number 1 S10 0D021508-01 for funding the Bruker Avance III 500 spectrometer. Protein Mass spectra were acquired by Dr. Maria Person, Dr. Stoney Herng-Hsiang Lo, Marvin Mercardo and Andre Bui in the Proteomics Facility at the University of Texas at Austin. High-resolution mass spectra were acquired by Dr. Yohannes H. Rezenom, in the Laboratory for Biological Mass Spectrometry, Department of Chemistry, Texas A&M University. Protein Nano-ESI spectra were acquired by Dr. Stephen J. Eyles, in the IALS Mass Spectrometry Core, Institute for Applied Life Sciences, University of Massachusetts Amherst. Protein NMR data were acquired at the CCNY NMR facility and the CUNY ASRC Biomolecular NMR Facility.

## Author contributions

T.S.K., W.H.J., N.D.E., A.P., D.Z.-O., P.R., R.G. and K.N.D. designed research; T.S.K., W.H.J., N.D.E., A.P., D.Z.-O., S.X.V.R., J.P., R.E., R.S., M.C., M.H., M.F.R., M.W. and K.Y.T. performed experiments; T.S.K., W.H.J., N.D.E., A.P., D.Z.-O., R.G., M.H., M.F.R., P.R., E.V.A., K.Y.T. and K.N.D. analyzed data, C.D.J.T. and J.P. contributed reagents/analytical tools. T.S.K. and K.N.D. wrote the paper, with input from other authors. All authors approved the final version of the manuscript.

## Competing interests

The authors declare no competing interests.
