## [Peer Review File · Nature Communications]

Reviewers' comments:

Reviewer #1 (Remarks to the Author): Expert in NMR and signal transduction

In this manuscript, the authors presented a small molecule BI-78D3 as an alternative inhibitor to suppress ERK activity. Overall, the authors provided a substantial amount of experimental data to demonstrate the therapeutic activity of BI-78D3 in treatment of melanoma cell lines and in vivo. This is an important topic, and the manuscript is well written. However, there is serious concern regarding target selectivity of BI-78D3s for ERK2, and whether this compound can form non-selective, covalent tagging to Cys residues exposed on the surface of different proteins. Several additional issues need to be addressed as follows.

1, The manuscript did not provide binding affinity of BI-78D3 to ERK2. BI-78D3 binds to the D-recruitment site (DRS) of ERK and forms covalent bond with Cys159, which is distal from the key pocket of ERK. However, the authors did not provide any data to show how well the compound and its analogs bind to the protein. Based on 2D NMR titration results, chemical perturbations of key residues are relatively small, except the gradual disappearance to Cys159 amide peak, indicating that the overall binding of the compound to the protein is rather weak.

2, Other than Cys159A, the authors did not attempt to make mutant residues in the ERK2 DRS pocket to confirm that BI-78D3 selectively interact into the DRS pocket. Based on ERK2 structure (PDB: 4erk), the DRS pocket is very small and shallow, does not seem suitable for a ligand binding. Indeed, from the modeling structure of BI-78D3 in complex with ERK2 (Fig 3b), I found that important functional groups of the compound do not fit into the pocket but stay outside of the DRS pocket, and the compound only forms a covalent bond with Cys159 that is on the protein surface and exposed to the solvent.

3, Authors show conserved sequences of DRS in many human MAPKs, but they did not offer any explanation why BI-78D3 only selectively targets DRS of ERK2. ERK2 protein has 6 Cys residues (PDB: 4erk), 4 Cys residues are buried in the protein internal structure and are well protected. Cys252 is on the protein surface and exposed to the outside, but Cys252 forms a hydrogen bond with Asn295 and is protected. Only Cys159 is exposed to the solvent and not protected, so it can readily react with BI-78D3 to form the co-valent bond. Additionally, JNK (PDB: 3v6r) protein also has 6 Cys residues, but they all seem to be buried inside the protein structure and protected. ERK5 (PDB: 5byy) has 3 Cys, which are not exposed to the solvent. Cys201 of JNK and Cys194 of ERK5 are corresponding to Cys159 of ERK2, but side chains of both residues do not seem exposed to the solvent, and are better protected by the protein surface than Cys159 of ERK2. Therefore, it is likely that BI-78D3 can work as a "tag" and react with many Cys residues on protein surfaces in cells, as long as they are exposed to the outside.

Reviewer #2 (Remarks to the Author): Expert in NMR and protein-ligand interactions

The manuscript by Kaoud et al. reports on a new approach to interfere with the activity of the ERK1 and ERK2 kinases by targeting the D-recruitment site (instead of the more commonly targeted ATP binding site) using BI-78D3, a small molecule that forms a covalent adduct with Cys-159.

Using NMR and other biophysical methods they convincingly show that BI-78D3 indeed forms a covalent adduct at the DRS in vitro.

They could also demonstrate the potency of this compound on cell culture and show that in vivo, BI-78D3 is able to inhibit ERK activation and cell proliferation.

I recommend the publication of the manuscript if the following points can be addressed:

- Since the BI-78D3 forms a covalent adduct with ERK1, Intermediate steps of the NMR based titrations should show 2 sets of resonances (instead of an averaged set of resonances e.g. in the

case of a low affinity non covalent binder). It would be nice to see an example of these intermediate steps were the 2 sets are be visible (for example in supp. figure 2)

- Along this line, the ¹⁵N TROSY based titration (figure 1C) shows that the intensity of the C159 crosspeak decreases upon addition of BI-78D3, which is expected if there is formation of a covalent adduct. What is more surprising is that the crosspeak also seems to experience chemical shift changes. I wonder how the author can explain these chemical shift changes. I'm also surprised by the choice of the authors to record this important experiment at 600 MHz where the benefit of the TROSY effect is marginal.

- The supp. Figure 7 is somewhat strange. If BI-78D3 is covalently attached to the kinase, it should experience the same correlation time as the protein. Meaning that the resonances of BI-78D3 should experience severe line broadening, which doesn't seem to be the case, at least for the resonance marked by an asterisk (that the authors depict as an evidence that the ligand is still attached to the protein). Additionally, according to the figure caption, the protein/ligand covalent complex went through a desalting step and many steps of buffer exchange, it is therefore surprising that the dioxane peak is still visible. Finally, the buffer is deuterated (according to the figure caption) but the amide resonances of the protein are still visible as well as an intense water peak. These discrepancies need to be clarified.

- Using NMR to verify the presence of the adduct after several washing steps of the protein is a good idea (as attempted in supp. Fig 7). The authors could incubate the ligand with uniformly deuterated protein and record the H-H NOESY spectrum of the attached ligand. Using an ILV sample they could even observe nOe crosspeaks between the protein and the ligand (which could help validate or improve the docking model).

Reviewer #3 (Remarks to the Author): Structure based design of inhibitors

The manuscript by Kaoud et al entitled: "Modulating multi-functional ERK complexes by covalent targeting of a recruitment site in vivo" presents data on a covalent ERK inhibitor, the 1,2,4-triazol-3-one BI-78D3, targeting the D-recruitment site (DRS). NMR data provided a convincing model of the interaction site which was confirmed by a site directed mutant (C159A) which rescued the anti-proliferative phenotype of this compound. The authors suggest a plausible mechanism of covalent attachment of BI-78D3 to ERK C159. The inhibitor has a $K_i = 2.3 \pm 0.8 \times 10^{-6}$ M and it is effective inhibiting colony growth in cell culture and also in mouse xenograft models.

Non-ATP competitive inhibitors of key pathways such as MAPK signalling are important tools for basic biology and future translational studies. The work is interesting and the study has been designed well. I support therefore publication of this paper. However, the authors should address the following issues in a revised version of the paper:

- Activity of the inhibitor is rather modest in cell culture and clearance is relatively fast. Are the authors confident that with a dosing of only 15 mg/kg BI-78D3 IP a sufficient exposure would be reached in vivo required for "on-target" inhibition? Have plasma levels of the drugs been monitored during this experiment. In cell based assays rather high concentration have been used (25 μ M).
- Covalent inhibitors usually have considerable activity for their designated targets which leads to high local concentration in the proximity of the residue targeted for bond formation. However, this does not seem to be the case for BI-78D3. In the pull down experiment two additional adducts have been observed (even though the level of incorporation seems low). Since this experiment monitors only bond formation with ERK if find it likely that BI-78D3 also reacts with other targets (some of them may influence MAPK signalling). A way to address this issue would be using a biotin labelled variant of the inhibitor in cellular pull down assays.

Minor issues: In the MS spectra of recombinant ERK the molecular weight of the adduct was 6 Da

larger than expected. How do the authors explain this mass difference (386 Da versus the expected 380 Da)

Reviewer #1 (Remarks to the Author): Expert in NMR and signal transduction

In this manuscript, the authors presented a small molecule BI-78D3 as an alternative inhibitor to suppress ERK activity. Overall, the authors provided a substantial amount of experimental data to demonstrate the therapeutic activity of BI-78D3 in treatment of melanoma cell lines and *in vivo*. This is an important topic, and the manuscript is well written. However, there is serious concern regarding target selectivity of BI-78D3 for ERK2, and whether this compound can form non-selective, covalent tagging to Cys residues exposed on the surface of different proteins. Several additional issues need to be addressed as follows.

We thank the reviewer for appreciating the importance of this study.

1. The manuscript did not provide binding affinity of BI-78D3 to ERK2. BI-78D3 binds to the D-recruitment site (DRS) of ERK and forms a covalent bond with Cys159, which is distal from the key pocket of ERK. However, the authors did not provide any data to show how well the compound and its analogs bind to the protein. Based on 2D NMR titration results, chemical perturbations of key residues are relatively small, except the gradual disappearance to Cys159 amide peak, indicating that the overall binding of the compound to the protein is rather weak.

We thank the reviewer and would like to take the opportunity to clarify our experiments. This was an issue we were acutely aware of and therefore spent some effort addressing mechanistically. Our data

support a mechanism where binding of BI-78D3 to ERK2 occurs through a two-step process of reversible binding, followed by a covalent step (*please see comment from reviewer #3 regarding this - 'The authors suggest a plausible mechanism of covalent attachment of BI-78D3 to ERK C159. The inhibitor has a $K_i = 2.3 \pm 0.8 \times 10^{-6} M$ '*). This is distinct from the 'tagging' mechanism reviewer #1 suggests because we show that the most efficient pathway for BI-78D3 to bind ERK covalently is through an initial 'recognition' process followed by a reaction step with the proximal cysteine. The best way to obtain an estimate for the equilibrium constant for this first step is through kinetic analysis of ERK2 inactivation by BI-78D3. This is indeed what we reported. Please note **Figure 3a**, which shows a plot of the observed rate constant for inactivation of ERK2 against the concentration of BI-78D3. At low concentrations of BI-78D3 (where the reversible binding is rate-limiting) the rate of inactivation is sensitive to its concentration, whereas at high concentrations the rate-limiting step is the chemical step. In this assay, activated ERK2 was incubated with different concentrations of BI-78D3 for varying amounts of time, before determining the residual activity of the enzyme that was not covalently bound. The line through the data corresponds to the best fit to eqn. ($k_{obs} = k_{inact} [I]/(K_i + [I])$) for a two-step mechanism of irreversible inhibition (shown in **Scheme 1, Figure 3a**), according to $K_i = 2.3 \pm 0.8 \times 10^{-6} M$ and $k_{inact} = 1.6 \pm 0.12 \times 10^{-3} s^{-1}$. Thus, the kinetic analysis suggests that BI-78D3 binds to the DRS of ERK2 with a K_i of approximately $2.3 \pm 0.8 \mu M$. Given the slow rate of covalent bond formation, the K_i is expected to be the same as the dissociation constant. We also present a displacement assay in Figure 1e, which measures the displacement of a fluorescent peptide from the DRS, which occurs with an approximate K_i of $0.8 \mu M$.

2. Other than Cys159A, the authors did not attempt to make mutant residues in the ERK2 DRS pocket to confirm that BI-78D3 selectively interact into the DRS pocket. Based on ERK2 structure (PDB: 4erk), the DRS pocket is very small and shallow, does not seem suitable for a ligand binding. Indeed, from the modeling structure of BI-78D3 in complex with ERK2 (Fig 3b), I found that important functional groups of the compound do not fit into the pocket but stay outside of the DRS pocket, and the compound only forms a covalent bond with Cys159 that is on the protein surface and exposed to the solvent.

We thank the reviewer and would like to provide some clarification. The molecular modeling was performed with a distant constraint of 5 kcal/A² and distance of 1.5 – 2.0 Å to sample the pre-reaction complex between the sulfur of C159 and the appropriate carbon atom of BI-78D3. Thus, the model may not represent the predominant binding mode for the reversible complex between ERK and BI-78D3. We have now provided clarification of this in the text.

Mutants	IC50 (μM)	K_m (μM)	k_{cat} (s ⁻¹)	k_{cat}/K_m (μM ⁻¹ s ⁻¹)
T108A	13.12±1.3	18.7±2.3	29.9±1.2	1.60
H123A	5.1±1.0	22.7±3.9	16.5±1.0	0.73
L155A	4.5±0.8	14.5±1.6	36.9±1.3	2.54
N156A	11.1±1.3	18.7±3.7	35.9±2.3	1.91
C159S	No inhib	12.8±1.9	17.3±0.8	1.35
C164A	3.1±0.6	9.30±0.9	19.6±0.5	2.10
D316A	4.4±0.7	20.7±5.0	17.4±1.4	0.84
D319A	5.4±0.8	17.2±1.9	12.1±0.4	0.70
WT	3.0±0.4	10.2±1.4	23.0±0.9	2.25

We have now mutated several residues to alanine (see Table above – **Supplementary Table 1**), that exhibited significant spectral perturbations upon adduct formation. Residues that were mutated include L155, N156, and C159 of loop 11, T108 of the inter-lobe linker, H123 of helix αE. Also, we mutated D316 and D319 of the Φ_{chg} pocket and C164 of the active site. Each mutant was expressed, purified, and activated and shown to possess steady-state parameters indistinguishable from the wild type enzyme. As expected, mutations at the Φ_{chg} pocket (D316A and D319A) and in the active site (C164A) did not affect the ability of BI-78D3 to inhibit ERK2 (determined as an apparent IC₅₀). Similarly, L155A, H123A were not significantly distinguishable from the wild type protein. However, mutation of C159S completely abrogated the ability of BI-78D3 to inhibit ERK2, while the T108A and N156A mutants showed a 3-4-fold increase in the apparent IC₅₀. These results, support the notion that BI-78D3 initially binds close to loop 11 (N156) and the spatially contiguous inter-lobe linker (T108) to facilitate formation of the covalent bond. Notably, these residues are unique to ERK1 and ERK2, suggesting that they may play a role in the specificity BI-78D3 shows towards them.

3. Authors show conserved sequences of DRS in many human MAPKs, but they did not offer any explanation why BI-78D3 only selectively targets DRS of ERK2. ERK2 protein has 6 Cys residues (PDB:

4erk), 4 Cys residues are buried in the protein internal structure and are well protected. Cys252 is on the protein surface and exposed to the outside, but Cys252 forms a hydrogen bond with Asn295 and is protected. Only Cys159 is exposed to the solvent and not protected, so it can readily react with BI-78D3 to form the covalent bond. Additionally, JNK (PDB: 3v6r) protein also has 6 Cys residues, but they all seem to be buried inside the protein structure and protected. ERK5 (PDB: 5byy) has 3 Cys, which are not exposed to the solvent. Cys201 of JNK and Cys194 of ERK5 are corresponding to Cys159 of ERK2, but side chains of both residues do not seem exposed to the solvent and are better protected by the protein surface than Cys159 of ERK2. Therefore, BI-78D3 can likely work as a “tag” and react with many Cys residues on protein surfaces in cells, as long as they are exposed to the outside.

We thank the reviewer for this insight. We hesitated to make conclusions regarding the reactivity of the cysteines from the structures, because of the uncertainty associated with the conformational dynamics of the proteins in solution. However, we agree that the structures do perhaps explain reduced activity in some instances. We have now attempted to assess the question experimentally, by comparing the reactivity of DTNB (which is similar in size to BI78D3, but highly reactive towards thiols and essentially serves as a 'tag') with BI-78D3. We prepared several recombinant MAPKs in phosphate buffer without reducing agent and allowed each to react with DNTB for 15 minutes. We also determined the ability of each MAPK to react with BI-78D3 (10 μ M for 60 minutes) by directly monitoring its UV-vis spectrum (please see the Figure to the right (and **Supplementary Figure 12**) – a peak between 300-400 nM corresponds to a reaction with BI-78D3). Besides, we determined the absolute mass of each MAPK, following incubation with BI-78D3 by LC-MS. The table below (and **Supplementary Table 2**) provides the moles of DTNB that react with each mole of a given MAPK. This reveals that all the MAPKs have at least one cysteine capable of reacting with DTNB. However, only ERK2 can react with BI-78D3. Interestingly, mutation of C163 of JNK2 (which corresponds to C159 in ERK2) to alanine did not impact the number of cysteines reactive with DNTB indicating, as the reviewer suggested, that access of BI-78D3 to this cysteine is restricted. We also looked at four additional proteins, BSA, esterase, aldolase A, and lysozyme. In no case did we observe any reactivity with BI-78D3.

We also compared the specificity of BI-78D3 and Ellman’s reagent (DNTB) towards C159 and C164 of ERK2 proteins. We generated four proteins, namely cysless ERK2 (all seven cys residues mutated), cysless ERK2 bearing only one cysteine at C159, cysless ERK2 bearing only one cysteine at C164 and cysless ERK2 bearing a cysteine at C159 and C164. Each of these was incubated with BI-78D3 for 60 minutes before reacting with Ellman’s reagent. DNTB readily reacted with both Cys (C159 and C164). BI-78D3 was able to protect C159 but not C164 from reacting with DNTB.

Protein Tested	DTNB mole/mole	Reported Cysteines	UV spectrum change with BI78D3
ERK2	3.6±0.4	7	Yes
JNK1	1.4±0.1	8	No
JNK2	0.7±0.05	10	No
JNK2-C163A	0.7±0.01	9	No
JNK3	1.0±0.06	11	No
p38alpha	3.0±0.1	4	No
ERK5 catalytic domain	1.9±0.4	4	No
BSA	0.4±0.06	35	No
Esterase	1.6±0.12	6	No
Aldolase A	1.5±0.11	3 ¹	No
Lysozyme	0.1±0.1	8 ²	No

¹one exhibits a fast reaction with DTNB; the other two are slow. ²all are in disulfide bonds.

Reviewer #2 (Remarks to the Author): Expert in NMR and protein-ligand interactions

The manuscript by Kaoud et al. reports on a new approach to interfere with the activity of the ERK1 and ERK2 kinases by targeting the D-recruitment site (instead of the more commonly targeted ATP binding site) using BI-78D3, a small molecule that forms a covalent adduct with Cys -159. Using NMR and other biophysical methods they convincingly show that BI-78D3 indeed forms a covalent adduct at the DRS in vitro. They could also demonstrate the potency of this compound on cell culture and show that in vivo, BI-78D3 can inhibit ERK activation and cell proliferation. I recommend the publication of the manuscript if the following points can be addressed:

We thank the reviewer for this encouraging comment.

1. Since BI-78D3 forms a covalent adduct with ERK1, intermediate steps of the NMR based titrations should show 2 sets of resonances (instead of an averaged set of resonances e.g., in the case of a low-affinity non-covalent binder). It would be nice to see an example of these intermediate steps where the 2 sets are visible (for example in supp. figure 2)

The reviewer is indeed correct; there are several examples of a second set of resonances (with increasing intensity) appearing with increasing concentration of ligand. We have now provided examples of two such sets of resonances in the new **Supplementary Figure 2**. Also see response to 2. below.

2. Along this line, the ^{15}N TROSY based titration (figure 1C) shows that the intensity of the C159 cross-peak decreases upon addition of BI-78D3, which is expected if there is formation of a covalent adduct. What is more surprising is that the cross-peak also seems to experience chemical shift changes. I wonder how the author can explain these chemical shift changes. I'm also surprised by the choice of the authors to record this important experiment at 600 MHz, where the benefit of the TROSY effect is marginal.

We want to thank the reviewer for this keen observation. The ligand was dissolved in DMSO, the presence of this solvent, not unexpectedly, had a small effect on some of the resonances. For this reason, the amide and methyl chemical shift perturbations shown in **Supplementary Figure 4a and 4b** were calculated utilizing an unbound reference containing the same amount of d_6 -DMSO (as the final point of the titration) in the absence of BI-78D3. **Figure 1C** and **Supplementary Figures 2 and 3a** show the raw titration data in which one sees the simultaneous effects of ligand binding and the introduction of DMSO in the buffer. It should be noted that the sole purpose of this TROSY-based titration was to confirm the expected quantitative disappearance of the C159 resonance. Indeed, upon addition of 10 mM DTT, we see the peaks reappear at positions consistent with the direction of the shifts noted by the reviewer. This is now illustrated in the new **Supplementary Figure 3b**.

The complete ^{15}N , ^1H TROSY-based titration set that is shown in **Figure 1c** and **Supplementary Figures 2 and 3** were performed to illustrate the quantitative modification of the resonance corresponding to C159. We feel that 600 MHz was sufficient for this purpose. However, as the reviewer will note, the Ile, Leu, Val methyl and amide chemical shift perturbations in ERK2 in the presence of an approximately equimolar amount of BI-78D3, where higher resolution (and sensitivity) was necessary, was performed at 800 MHz. These data are shown in **Supplementary Figure 4** along with representative spectra.

3. The supp. Figure 7 is somewhat strange. If BI-78D3 is covalently attached to the kinase, it should experience the same correlation time as the protein. Meaning that the resonances of BI-78D3 should experience severe line broadening, which doesn't seem to be the case, at least for the resonance marked by an asterisk (that the authors depict as an evidence that the ligand is still attached to the protein).

We agree with the reviewer's concern regarding the resonance marked by an asterisk in the Supplementary Figure 7b (now **Supplementary Figure 10a**); this indeed requires further clarification. The S-moiety of BI-78D3 is extremely flexible, and when BI-78D3 is covalently attached to ERK2, this moiety is expected to be surface exposed. This increases its chance to freely rotate in the solvent, explaining the sharpness of the proton peak in Supplementary Figure 7b (now **Supplementary Figure 10a**). This is somewhat analogous to the fact that S^2_{axis} values of Met ϵ -methyls are the lowest among all methyl-bearing groups. When we expanded the resonance marked by the asterisk in the Supplementary Figure 7b (now **Supplementary Figure 10a**) (see Figure to the right and new **Supplementary Figure 10b**), we noted that the proton peak at 8.3 ppm of the ERK2-BI-78D3 (6) is broader and shifted downfield in comparison to the same peak in an authentic sample of BI-78D3 (1) (full width at half maximum ~ 2.0 and 1.22 Hz respectively). Note a similar chemical shift was observed for the same proton when BI-78D3 was reacted with one equivalent of methoxy ethanethiol (Supplementary Figure 7a, (now **Supplementary Figure 9a**)).

Additionally, according to the figure caption, the protein/ligand covalent complex went through a desalting step and many steps of buffer exchange, it is therefore surprising that the dioxane peak is still visible. Finally, the buffer is deuterated (according to the figure caption) but the amide resonances of the protein are still visible as well as an intense water peak. These discrepancies need to be clarified.

We want to thank the reviewer for emphasizing concerns regarding this experiment. Initially, the NMR experiment was performed twice. The NMR of the first experiment was run in 0.6 mL of phosphate buffer (H_2O) with 30 μ L dioxane- d_8 . In this experiment the assigned proton peak at 8.3 was present, but so was a large peak from water. Therefore, in the second experiment, which we showed in the manuscript (Supplementary Figure 5), after the reaction was completed, and the desalting step was performed using a PD10 column, we exchanged the phosphate buffer (H_2O) with 50 mM phosphate buffer (D_2O , without dioxane) six times, using a 1:2 ratio (effectively 64-fold). Some water and dioxane was still present due to the partial exchange of water and dioxane. However, the resolution was improved over the first experiment. The protein-ligand interaction was confirmed in this experiment using mass spectrometry (**Figure 2c**). In a third experiment performed for this revision, after completing the reaction and the desalting step, the PD10 column fractions containing the protein were combined, concentrated and the buffer exchanged 11 times, each by 1:3 dilution in deuterated buffer without dioxane. Interestingly, the NMR spectra of this experiment still showing the dioxane peak even after excessive exchange of the buffer. In all three experiments, the protein-ligand interaction was monitored by UV spectroscopy. In each case, the characteristic shift of the UV spectrum of the ligand indicating covalently attachment to ERK2 (**Figure 2b**) was observed, further confirming the integrity of the experiments.

4. Using NMR to verify the presence of the adduct after several washing steps of the protein is a good idea (as attempted in supp. Fig 7). The authors could incubate the ligand with uniformly deuterated protein and record the H-H NOESY spectrum of the attached ligand. Using an ILV sample, they could even observe noe cross-peaks between the protein and the ligand (which could help validate or improve the docking model).

We want to thank the reviewer for this suggestion. We have provided many orthogonal pieces of evidence to demonstrate that BI-78D3 is indeed covalently attached to C159 including an LC-MS analysis and UV-vis spectroscopy. We do not feel that an additional confirmatory NMR experiment will add to this.

We also do not think, given the lack of Ile, Val and Leu methyls in the immediate vicinity of the attached BI-78D3, and the fact that only a single position (marked by a '*' and discussed above) of the attached BI-78D3 molecule provides signal with high sensitivity, that meaningful NOEs between the attached BI-78D3 and ILV-labeled ERK2 will be obtained. While a high-resolution structure of the adduct is something we will pursue in the short-term, this is best done using X-ray crystallography, though there are questions whether density may be visible for the entire dynamic BI-78D3 molecule.

Reviewer #3 (Remarks to the Author): Structure-based design of inhibitors

The manuscript by Kaoud et al. entitled: "Modulating multi-functional ERK complexes by covalent targeting of a recruitment site in vivo" presents data on a covalent ERK inhibitor, the 1,2,4-triazol-3-one BI-78D3, targeting the D-recruitment site (DRS). NMR data provided a convincing model of the interaction site, which was confirmed by a site-directed mutant (C159A) which rescued the anti-proliferative phenotype of this compound. The authors suggest a plausible mechanism of covalent attachment of BI-78D3 to ERK C159. The inhibitor has a $K_i = 2.3 \pm 0.8 \times 10^{-6}$ M, and it is effective inhibiting colony growth in cell culture and also in mouse xenograft models. Non-ATP competitive inhibitors of key pathways such as MAPK signaling are important tools for basic biology and future translational studies. The work is interesting, and the study has been designed well. I support, therefore, publication of this paper. However, the authors should address the following issues in a revised version of the paper:

We want to thank the reviewer for this encouraging comment.

1. Activity of the inhibitor is rather modest in cell culture, and clearance is relatively fast. Are the authors confident that with a dosing of only 15 mg/kg BI-78D3 IP a sufficient exposure would be reached in vivo required for "on-target" inhibition? Have plasma levels of the drugs been monitored during this experiment? In cell-based assays, rather high concentration have been used (25 μ M).

We appreciate the reviewer's commentary on the in vivo potency of BI-78D3. We show in **Supplementary Figure 24** (to the right here) that cells expressing wild type BRAF and KRAS (e.g., HEK293 cells), as well as mutant KRAS (e.g., A549 cells) are significantly more resistant to BI-78D3 than mutant BRAF-expressing cells (e.g. A375 cells), and in all cases resistance appears to correlate with ERK inhibition. BI-78D3 is reported to have a plasma half-life of almost 60 minutes (42% remains in the plasma after 60 minutes)¹, and herein we established its in-vivo effect on ERK phosphorylation and tumor growth using A375 BRAF-mutant xenograft model (**Figure 5a and 5b**). Suggesting its ability to stay intact without depletion by cellular glutathione until it reaches its target on the surface of ERK and form the covalent adduct. Previously, Cheung et al. tested BI-78D3 in OVK18 xenografts that have a naturally occurring PIK3R1_{L370fs} mutant. OVK18 cells were injected subcutaneously into the mice flank region (n=5/group), and treatment with BI-78D3 (10 mg/kg) was administered i.p. 4 times per week for 3 weeks.² This dose was enough to inhibit the tumor growth if compared to the group that were treated by the vehicle control. In this report, the authors did not identify ERK as a target of BI-78D3, but 10 mg/kg dosage was enough to inhibit the tumor growth in this xenograft model.

2. Covalent inhibitors usually have considerable activity for their designated targets, which leads to high local concentration in the proximity of the residue targeted for bond formation. However, this does not seem to be the case for BI-78D3. In the pull-down experiment, two additional adducts have been observed (even though the level of incorporation seems low). Since this experiment monitors only bond formation with ERK if find it likely that BI-78D3 also reacts with other targets (some of them may influence MAPK signaling). A way to address this issue would be using a biotin-labeled variant of the inhibitor in cellular pull-down assays.

We thank the reviewer for initiating this interesting discussion, and we agree with the reviewer that the adducts that are shown in the mass spectra of **Figure 4a** require further clarification. This experiment was designed to examine the successful targeting of ERK2 by BI-78D3 in mammalian cells. HEK293 cells in which flag-ERK2-WT was overexpressed, were treated with BI-78D3 for 2 hours, cells were lysed and flag-ERK2 purified using flag-beads (Sigma). The purification process was achieved in around 90-120 minutes (methods section), and flag-ERK2 was eluted using 3x flag peptide (Sigma). Purified flag-ERK2 was treated with DMSO or BI-78D3, flash frozen to -80 and sent for mass spectrometry studies. The deconvoluted spectrum of flag-ERK2 from BI-78D3-treated cells displayed mass shifts of 386 Da, suggesting the addition of one molecule of BI-78D3 to each molecule of flag-ERK2. All the adducts purified in this experiment and shown in the spectra represent different forms of ERK2, as the flag beads are expected to pull down flag-ERK2 specifically.

The mass spectrum of the untreated cells (**Figure 4a, upper panel**), showed three forms of ERK2, which we attribute to unphosphorylated (42450), mono-phosphorylated (42524) and bi-phosphorylated (42589) flag-ERK2. The BI78-D3-treated cells show three peaks corresponding to each form of flag-ERK2 labeled by BI-78D3 (**Figure 4a, bottom panel**)

	Calc. mass of Flag-tagged ERK2	Observed forms of ERK2 in the non-treated cells (Fig 4a Upper)	Calc. mass of each observed form of ERK2 after BI78D3 treatment +380	Observed ERK2 adducts in the treated cells (Fig 4a Lower)	Difference between calc. and obs. adducts in the treated cells
Unphosphorylated Flag-ERK2	42401	42450 (49)	42450+380=42830	42836	+6
mono-phosphorylated Flag-ERK2	42481	42524 (43)	42524+380=42903	42909	+5.4
Bi-phosphorylated Flag-ERK2	42561	42589 (28)	42589+380=42968	42960	-8.6

The masses of each phosphorylation state of ERK2 in the un-treated cells exhibits a small shift from the calculated masses (the first two columns in the above Table). We attribute these differences to the presence of some contaminants which appeared during the process. For example, small amounts of acetonitrile (42 Da), formic acids (44 Da) or acetic acids (59 Da) could have been picked up during the overall process. Also, acetylation of ERK2 in mammalian cells or association of ERK2 with metals (e.g., iron) can produce a mass shift. In conclusion, obtaining mass spectra of endogenous proteins is challenging, and while excessive purification of the samples may help decrease the level contaminants, we have to balance this with the potential for losing material.³

Despite the small differences between the expected and observed masses, we are confident that all the masses observed in this experiment correspond to one of the three forms of ERK2 and that the three ERK2 adducts observed in the treated cells correspond to 100% labeling of ERK2 by BI78D3. None of these adducts represent any off-target protein of BI-78D3.

In this revision, we examined the possibility that BI-78D3 would interact with other cellular proteins carrying a solvent exposed cysteine. We tested several MAP kinases and four random proteins. We showed in our response to reviewer 1, point 3 that all the proteins we tested contain at least one cysteine that can freely react with DTNB. However, BI-78D3 only reacts with Cys159 of ERK2. We consider these experiments to be sufficient to address the fundamental specificity of BI-78D3 and certainly do not rule out the possibility that BI-78D3 forms adducts with some other cellular proteins. Our plan moving forward is to develop a molecular probe to examine the specificity of BI-78D3 and analogs, with the caveat in mind that any probe molecule will be expected to have a different specificity.

3. Minor issues: In the MS spectra of recombinant ERK, the molecular weight of the adduct was 6 Da larger than expected. How do the authors explain this mass difference (386 Da versus the expected 380 Da)

We thank the reviewer for pointing this out, initially we have asked Dr. Maria Person (The director of the proteomics facility in the college of pharmacy here in UT) about the accuracy of the instruments we used to generate most of these mass spectra data in the manuscript, she implied that the acceptable error for a 50 kDa protein is around ± 10 Da. It appears to be generally accepted that the current mass accuracy of many ESI interfaced commercial mass spectrometry is around 0.01% or ± 5 Da for 50 kDa protein.⁴

References

- 1 De, S. K. *et al.* Synthesis and optimization of thiadiazole derivatives as a novel class of substrate competitive c-Jun N-terminal kinase inhibitors. *Bioorg Med Chem* **18**, 590-596, doi:10.1016/j.bmc.2009.12.013 (2010).
- 2 Cheung, L. W. *et al.* Naturally occurring neomorphic PIK3R1 mutations activate the MAPK pathway, dictating therapeutic response to MAPK pathway inhibitors. *Cancer Cell* **26**, 479-494, doi:10.1016/j.ccell.2014.08.017 (2014).
- 3 Gingras, A. C., Aebersold, R. & Raught, B. Advances in protein complex analysis using mass spectrometry. *J Physiol* **563**, 11-21, doi:10.1113/jphysiol.2004.080440 (2005).
- 4 Pramanik, B. N., Lee, M. S., Chen, G. & Wiley InterScience (Online service). In *Wiley series on pharmaceutical science and biotechnology*. 1 online resource (xxii, 462 pages) (John Wiley, Hoboken, N.J., 2011).

REVIEWERS' COMMENTS:

Reviewer #1 (Remarks to the Author):

The authors made excellent efforts in the revision of this manuscript. They have successfully addressed my previous comments by clarification and providing additional experimental data. The revised manuscript is recommended for publication.

Reviewer #2 (Remarks to the Author):

I'm satisfied with the changes and answers provided by the authors and recommend the publication of the manuscript.

Reviewer #3 (Remarks to the Author):

The authors responded adequately to the issues that I raised – I have therefore no further concerns regarding publication of this manuscript in Nat. Comm.

Kind regards

Stefan Knapp, University of Frankfurt